# Calibration by Distribution Matching: Trainable Kernel Calibration Metrics

**Charles Marx**[*]
Stanford University
ctmarx@cs.stanford.edu

**Sofian Zalouk**[*]
Stanford University
szalouk@stanford.edu

**Stefano Ermon**
Stanford University
ermon@cs.stanford.edu

## Abstract

Calibration ensures that probabilistic forecasts meaningfully capture uncertainty by requiring that predicted probabilities align with empirical frequencies. However, many existing calibration methods are specialized for post-hoc recalibration, which can worsen the sharpness of forecasts. Drawing on the insight that calibration can be viewed as a distribution matching task, we introduce kernel-based calibration metrics that unify and generalize popular forms of calibration for both classification and regression. These metrics admit differentiable sample estimates, making it easy to incorporate a calibration objective into empirical risk minimization. Furthermore, we provide intuitive mechanisms to tailor calibration metrics to a decision task, and enforce accurate loss estimation and no regret decisions. Our empirical evaluation demonstrates that employing these metrics as regularizers enhances calibration, sharpness, and decision-making across a range of regression and classification tasks, outperforming methods relying solely on post-hoc recalibration.

## 1 Introduction

Probabilistic forecasts are valuable tools for capturing uncertainty about an outcome of interest. In practice, the exact outcome distribution is often impossible to recover [51], leading to overconfident forecasts [15]. Calibration provides statistical guarantees on the forecasts, ensuring that the predicted probabilities align with the true likelihood of events. For example, consider a weather forecast that assigns an 80% probability of rain tomorrow. A well-calibrated forecast would imply that, on average, it rains on 80% of the days with such predictions. By appropriately quantifying uncertainty, calibration empowers decision-makers to efficiently allocate resources and mitigate risks, even when the outcome distribution cannot be recovered exactly.

Many forms of calibration have been proposed in both classification [8, 21, 35] and regression [42, 23, 53] to ensure that uncertainty estimates match true frequencies. From a methodological perspective, the literature on enforcing calibration is fractured; many algorithms to enforce calibration are specialized for a particular form of calibration [39, 29, 52, 10, 17] or are designed exclusively for post-hoc calibration [27, 23]. Although post-hoc methods are effective at enforcing calibration, they often lead to a degradation in the predictive power, i.e. sharpness, of the forecaster [26]. This motivates the need for versatile metrics that can enforce various forms of calibration during training.

In this work, we introduce a unified framework wherein existing notions of calibration are presented as distribution matching constraints. To enforce distribution matching, we use the Maximum Mean Discrepancy (MMD), giving us unbiased estimates of miscalibration that are amenable to gradient-based optimization. We frame these metrics as regularizers and optimize them alongside proper scoring rules. This allows us to enforce calibration while preserving sharpness.

---

[*]Equal Contribution.

37th Conference on Neural Information Processing Systems (NeurIPS 2023).

Framing calibration as distribution matching [38] allows us to use the Maximum Mean Discrepancy (MMD) [14] to give unbiased estimates of miscalibration that are amenable to gradient-based optimization. An advantage of this approach is that we express existing calibration measures by varying the kernel in the MMD metric. Moreover, we can express new notions of calibration specific to a downstream decision task. This allows us to prioritize errors that impact decision-making [52]. Our contributions can be summarized as follows:

- We propose a framework based on distribution matching to unify many notions of calibration.

- We provide differentiable training objectives to incentivize a wide variety of popular forms of calibration during empirical risk minimization in supervised learning. We frame these metrics as regularizers that can be optimized alongside proper scoring rules. This allows us to enforce calibration without degrading sharpness.

- We show how to define and enforce new calibration metrics tailored to specific downstream decision problems. When these metrics are successfully optimized, decision makers can accurately estimate the decision loss on unlabeled data.

- We perform an empirical study across 10 tabular prediction tasks (5 regression and 5 classification) and find that using our metrics as regularizers improves calibration, sharpness, and decision-making relative to existing recalibration methods.

## 2  Related Work

**Calibration**  Our work builds on the literature on calibrated forecasting [31, 8, 9, 13]. Recently, many forms of calibration have been proposed in classification [5, 25, 24, 32, 35, 33, 10, 17, 28, 36, 27], regression [23, 42, 53, 39, 51], and beyond [47, 22]. Algorithms to enforce calibration are often designed specifically for one form of calibration. For example, significant attention has been devoted to effectively estimating the Expected Calibration Error (ECE) to calibrate confidence levels in binary classification [19, 44, 4, 15, 35]. Our work aims to supplement this work by providing general calibration metrics that can be used as training objectives. These methods can be used in concert with post-hoc calibration, to encourage calibration during model training and correct any remaining miscalibration after training is complete.

**Decision Calibration**  An interesting line of recent work focuses on calibrating predictions for downstream decision problems [39, 52]. Zhao et al. [52] shows that decision calibration can be achieved when the set of possible actions is finite. We show cases in which this result can be extended to infinite action spaces by leveraging a low-dimensional sufficient statistic of the decision problem. Whereas Zhao et al. [52] compute calibration by approximating a worst-case decision problem with a linear classifier, we use the kernel mean embedding form of an MMD objective, making it simple to calibrate with respect to a restricted set of loss functions by adjusting the kernel.

**Kernel-Based Calibration Metrics**  Motivated by issues arising from the histogram binning of the Expected Calibration Error, a recent stream of research uses kernels to define smooth calibration metrics. Zhang et al. [50] and Popordanoska et al. [37] use kernel density estimates to estimate calibration in classification settings. In contrast, our work applies to regression and takes a distribution matching perspective. Widmann et al. [47] and Kumar et al. [26] are the most similar existing works to our own, and also introduce MMD-based calibration metrics. Our work differs from Widmann et al. [46] due to our focus on trainability and optimization, connections to existing forms of calibration, and applications to decision-making. Kumar et al. [26] can be seen as a special case of our method applied to top-label calibration in multiclass classification (see Table 1). Our work extends the literature on MMD-based calibration metrics, showing how to express and optimize 11 existing forms of calibration by varying the kernel in the MMD objective (see Tables 1 and 2).

## 3  Calibration is Distribution Matching

We consider the regression setting, where, given a feature vector $x \in \mathcal{X} = \mathbb{R}^d$, we predict a label $y \in \mathcal{Y} \subset \mathbb{R}$. We are given access to $n$ i.i.d. examples $(x_1, y_1), \ldots, (x_n, y_n)$ distributed according to some true but unknown cumulative distribution function (cdf) $P$ over $\mathcal{X} \times \mathcal{Y}$. A lower case $p$ denotes the probability density function (pdf) or probability mass function (pmf), when it exists. We use

Table 1: We express popular forms of calibration in terms of distribution matching, where $P$ is the true distribution and $Q$ is the forecast. The forecaster achieves calibration if the forecast variable and target variable are equal in distribution, given the conditioning variable. Here, $\delta_\ell^*(Q_{Y|X})$ is a Bayes optimal action under $Q_{Y|X}$, $y_0$ is a fixed label in $\mathcal{Y}$, $\alpha \in [0, 1]$ is a threshold, and $\phi(X)$ is a feature mapping. See Appendix A for additional details.

| Calibration Objective | Forecast Variable | Target Variable | Conditioning Variable |
|---|---|---|---|
| Quantile Calibration [23] | $Q_{Y|X}(Y)$ | $P_{Y|X}(Y)$ | − |
| Threshold Calibration [39] | $Q_{Y|X}(Y)$ | $P_{Y|X}(Y)$ | $\mathbb{1}\left\{Q_{Y|X}(y_0) \leq \alpha\right\}$ |
| Marginal Calibration [12] | $\widehat{Y}$ | $Y$ | − |
| Decision Calibration [52] | $\widehat{Y}$ | $Y$ | $\delta_\ell^*(Q_{Y|X})$ |
| Group Calibration [36] | $\widehat{Y}$ | $Y$ | $\mathrm{Group}(X)$ |
| Distribution Calibration [42] | $\widehat{Y}$ | $Y$ | $Q_{Y|X}$ |
| Individual Calibration [51] | $\widehat{Y}$ | $Y$ | $X$ |
| Local Calibration [27] | $\widehat{Y}$ | $Y$ | $\phi(X)$ |

subscripts for marginal and conditional distributions, so $P_{Y|x}(\cdot)$ is the conditional cdf for $Y$ given $X = x$ and $p_Y(\cdot)$ is the marginal pdf for $Y$. When the input $X$ is random, we write $P_{Y|X}$ as the cdf-valued random variable that takes value $P_{Y|x}$ upon the event that $X = x$.

Our goal is to learn a forecaster $Q$ that, given a feature vector $x$, forecasts a cdf $Q_{Y|x}(\cdot)$ over $\mathcal{Y}$. A forecaster implicitly defines a joint distribution over $\mathcal{X} \times \mathcal{Y}$ by combining the forecasted conditional pdf with the marginal pdf for the features, i.e. $q(x, y) = p_X(x)q_{Y|x}(y)$. The forecaster's joint cdf $Q(x, y)$ is defined by integrating the pdf $q(x, y)$ over the appropriate region of $\mathcal{X} \times \mathcal{Y}$. In the same way as $(X, Y) \sim P$ is a sample from the true distribution, we write a sample from $Q$ as $(X, \widehat{Y}) \sim Q$, where $X \sim P_X$ and $\widehat{Y}$ is a label drawn from the forecast $(\widehat{Y}|X = x) \sim Q_{Y|x}$.

An ideal forecaster perfectly recovers the true probabilities so that $Q = P$. However, in practice we often only observe each feature vector $x$ once, making perfect forecasts unattainable [51]. Instead, calibration enforces that forecasts are accurate *on average*, leaving us with a tractable task that is strictly weaker than perfect forecasting. Each form of calibration requires the forecasts to respect a particular law of probability. For example, if $Y$ is binary, distribution calibration states that on examples for which the forecast is $q_{Y|x}(Y = 1) = c$, we should observe that $Y = 1$ with frequency $c$. In regression, quantile calibration [23] states that $y$ should exceed each quantile $c \in [0, 1]$ of the forecast with frequency $c$. Calibration can also be enforced within subgroups of the data (i.e. group calibration [36]), in smooth neighborhoods of the input space (i.e. local calibration [27]), and even for a single feature vector (i.e. individual calibration [51]) by using randomization.

Calibration can be expressed as a conditional distribution matching problem, requiring that certain variables associated with the true distribution match their counterparts from the forecasts.

**Lemma 3.1** (informal). *Quantile calibration is equivalent to distribution matching between $Q_{Y|X}(Y)$ and $P_{Y|X}(Y)$. Distribution calibration is equivalent to distribution matching between $\widehat{Y}$ and $Y$ given $Q_{Y|X}$. Individual calibration is equivalent to distribution matching between $\widehat{Y}$ and $Y$ given $X$.*

The choice of random variables that we match in distribution determines the form of calibration recovered. We show how to express many popular forms of calibration in terms of distribution matching in Table 1, and include additional details on these correspondences in Appendix A. In Section 4, we describe how to express and estimate these forms of calibration using kernel-based calibration metrics. In Section 5, we show how to tailor these metrics to downstream decision problems. Finally, in Section 6, we apply these metrics as regularizers and explore the empirical effects on calibration, sharpness, and decision-making.

## 4   Calibration Metrics as Trainable Regularizers

In this section, we introduce a framework to enforce calibration using distribution matching. We quantify calibration using integral probability metrics which admit unbiased estimates from data

and are amenable to gradient-based optimization. This allows us to optimize calibration alongside a standard training objective, achieving calibration without degrading sharpness. Jointly optimizing calibration and sharpness has been shown to outperform only enforcing calibration post-hoc [26].

Integral probability metrics (IPMs) are a flexible set of metrics to quantify the difference between two distributions [30] in terms of a family of witness functions. Since we need to perform distribution matching *conditionally*, we consider witness functions over both the label $Y$, as well as a conditioning variable $Z$ that assumes the same distribution under $P$ and $Q$.

$$D(\mathcal{F}, P, Q) = \sup_{f \in \mathcal{F}} \left| \mathbb{E}_P \left[ f(Y, Z) \right] - \mathbb{E}_Q \left[ f(\widehat{Y}, Z) \right] \right|. \tag{1}$$

Different choices for the witness functions $\mathcal{F}$ and the conditioning variable $Z$ measure different notions of calibration. Similar to Widmann et al. [47], we focus on the case where $\mathcal{F}$ is the unit ball of a Reproducing Kernel Hilbert Space (RKHS) $\mathcal{H}$, so that the IPM corresponds to the Maximum Mean Discrepancy [14], denoted $\mathrm{MMD}(\mathcal{F}, P, Q)$. MMD can either be expressed in terms of the canonical feature map $\phi(y, z) \in \mathcal{H}$ or the reproducing kernel $k((y, z), (\widehat{y}, z')) = \langle \phi(y, z), \phi(\widehat{y}, z') \rangle_{\mathcal{H}}$ for $\mathcal{H}$:

$$\mathrm{MMD}^2(\mathcal{F}, P, Q) = \left\| \mathbb{E}_P \left[ \phi(Y, Z) \right] - \mathbb{E}_Q \left[ \phi(\widehat{Y}, Z) \right] \right\|_{\mathcal{H}}^2 \tag{2}$$

$$= \mathbb{E}_P \left[ k((Y, Z), (Y', Z')) \right] + \mathbb{E}_Q \left[ k((\widehat{Y}, Z), (\widehat{Y}', Z')) \right] - 2 \mathbb{E}_P \mathbb{E}_Q \left[ k((Y, Z), (\widehat{Y}', Z')) \right]$$

Given a dataset $D$ consisting of $n$ i.i.d. pairs $(x_i, y_i) \sim P$, we can generate forecasted labels $\widehat{y}_i \sim Q_{Y|x_i}$. The conditioning variables $z_i$ are computed from $(x_i, y_i)$ according to the expressions given in Table 1. The plug-in MMD estimator is unbiased and takes the form [14]:

$$\widehat{\mathrm{MMD}}^2(\mathcal{F}, D, Q) = \frac{1}{n(n-1)} \sum_{i=1}^{n} \sum_{j \neq i}^{n} h_{ij}, \tag{3}$$

where $h_{ij}$ is a one-sample U statistic:

$$h_{ij} := k((y_i, z_i), (y_j, z_j)) + k((\widehat{y}_i, z_i), (\widehat{y}_j, z_j)) - k((y_i, z_i), (\widehat{y}_j, z_j) - k((y_j, z_j), (\widehat{y}_i, z_i))$$

The variance of this estimator can be reduced by marginalizing out the simulated randomness used to sample $\widehat{y}_i \sim Q_{Y|x_i}$. This can be done either analytically, or empirically by resampling the forecasted labels multiple times per example. We find that empirically marginalizing out the simulated randomness improves training stability in practice. The full training objective combines a proper scoring rule (we use negative log-likelihood) with an MMD estimate to incentivize calibration. To train, we divide the dataset $D$ into batches $D_b \subset \{1, \ldots, n\}$ large enough so that the MMD estimate on a batch has low enough variance to provide useful supervision (we find that $D_b$ of size 64 suffices in practice). Given a family of predictive models $\{Q_\theta : \theta \in \Theta\}$ (e.g., a neural network), we optimize the following training objective on each batch:

$$\min_{\theta \in \Theta} \sum_{i \in D_b} -\log q_{Y|x_i;\theta}(y_i) + \lambda \cdot \widehat{\mathrm{MMD}}^2(\mathcal{F}, D_b, Q_\theta) \tag{4}$$

In order to differentiate $\widehat{\mathrm{MMD}}^2(\mathcal{F}, D_b, Q_\theta)$ with respect to $\theta$, we require that the kernel $k$ is differentiable and the samples can be expressed as a differentiable function of the model parameters (e.g., using a reparameterization trick). For parameterizations that do not admit exact differentiable sampling, we can apply a differentiable relaxation (e.g., if $Q_\theta$ is a mixture distribution, we can use a Gumbel-Softmax approximation). In principle, it is possible to remove the need for differentiable sampling altogether by optimizing our MMD objective using Monte-Carlo RL techniques (e.g., REIN-FORCE [48]). However, this is beyond the scope of our work. Note that regularizing for calibration at training time does not give calibration guarantees, unless one can guarantee the objective will be optimized out-of-sample. In this sense, regularization and post-hoc calibration are complementary: regularization mitigates the trade-off between calibration and sharpness while post-hoc methods can provide finite-sample calibration guarantees (e.g., [45, 29]). For this reason, we suggest using regularization and post-hoc calibration together (see Table 3).

**On the Choice of the Kernel** When the kernel $k$ is universal, the MMD metric is zero if and only if $Y$ and $\widehat{Y}$ are equal in distribution, given $Z$ [14]. Universality is preserved when we decompose the kernel over $\mathcal{Y} \times \mathcal{Z}$ into a pointwise product of two universal kernels, one over $\mathcal{Y}$ and one over $\mathcal{Z}$ [43].

Table 2: Popular forms of calibration expressed as distribution matching for a classification problem with classes $\mathcal{Y} = \{1, \ldots, m\}$. Recall that $q_{Y|X}(\cdot)$ is the forecasted pmf. The forecaster is calibrated if the forecast variable and target variable are equal in distribution, given the conditioning variable. For top-label calibration, we use $Y^* := \arg\max_{y \in \mathcal{Y}} q_{Y|X}(y)$, which is random due to $X$. The marginal calibration condition expresses $m$ distribution matching constraints, one for each $y \in \mathcal{Y}$.

| Calibration Objective | Forecast Variable | Target Variable | Conditioning Variable |
|---|---|---|---|
| Canonical Calibration [44, Eq 1] | $\widehat{Y}$ | $Y$ | $q_{Y|X}$ |
| Top-label Calibration [44, Eq 2] | $\mathbb{1}\{\widehat{Y} = Y^*\}$ | $\mathbb{1}\{Y = Y^*\}$ | $q_{Y|X}(Y^*)$ |
| Marginal Calibration [44, Eq 3] | $\mathbb{1}\{\widehat{Y} = y\}$ | $\mathbb{1}\{Y = y\}$ | $q_{Y|X}(y), \forall y \in \mathcal{Y}$ |

**Lemma 4.1.** *Let $k$ be the function defined by the pointwise product $k((y, z), (y', z')) = k_y(y, y')k_z(z, z')$ where $k_y$ and $k_z$ are universal kernels over spaces $\mathcal{Y}$ and $\mathcal{Z}$ respectively. Then, under mild conditions, $k$ is a universal kernel over $\mathcal{Y} \times \mathcal{Z}$.*

See [43, Theorem 4] for a proof. In our experiments, we explore using a universal RBF kernel over $\mathcal{Y} \times \mathcal{Z}$, since it satisfies the mild conditions needed for Lemma 4.1. To enforce decision calibration, we also consider a kernel $k_Y$ that is intentionally not universal, and instead only measures differences between $P$ and $Q$ that are relevant for decision-making (see Section 5). Furthermore, to measure Quantile Calibration and Threshold Calibration, we must match the distributions of the probability integral transforms $P_{Y|X}(Y)$ and $Q_{Y|X}(Y)$ instead of $Y$ and $\widehat{Y}$. Thus, we replace $k_Y$ with a universal kernel over $[0, 1]$, the codomain of the probability integral transform.

## 4.1 Calibration Metrics for Classification

Thus far, we have focused on estimating calibration metrics in the regression setting. We now describe how our framework applies to classification.

In the classification setting, given a feature vector $x \in \mathcal{X} = \mathbb{R}^d$ we predict a label $y \in \mathcal{Y} = \{1, \ldots, m\}$. As before, we are given access to $n$ i.i.d. examples $(x_1, y_1), \ldots, (x_n, y_n)$ distributed according to some unknown distribution $P$ over $\mathcal{X} \times \mathcal{Y}$. Our goal is to learn a forecaster $Q$ that, given a feature vector $x$, forecasts a probability mass function (pmf) $q_{Y|x}(\cdot)$ over labels $\mathcal{Y}$.

We express popular notions of calibration used in classification (i.e., canonical, top-label, marginal) in terms of distribution matching in Table 2. See Appendix A.2 for additional details on calibration and distribution-matching in classification. We use the same unbiased plug-in estimator for MMD shown in Equation 3. In the classification setting, we can analytically marginalize out the randomness originating from the forecast when computing the MMD estimate. After performing this marginalization, the one-sample U-statistic $h_{ij}$ is is given by

$$h_{ij} = k((y_i, z_i), (y_j, z_j)) + \sum_{y \in \mathcal{Y}} \sum_{y' \in \mathcal{Y}} q_i(y)q_j(y')k((y, z_i), (y', z_j)) - 2 \sum_{y \in \mathcal{Y}} q_i(y)k((y, z_i), (y_j, z_j)),$$

where the conditioning variables $z_i$ are computed from $(x_i, y_i)$, as detailed in Table 2. Here, $q_i(y) = q_{Y|x_i}(y)$ is the predicted probability of the label $Y = y$ given features $x_i$. When the forecast and target variables depend on variables other than $Y$ (e.g., $Y^*$ depends on $q_{Y|X}$ in top-label calibration), we add those variables as inputs to the kernel. Note that the plug-in MMD estimator with this U-statistic is differentiable in terms of the predicted label probabilities, and that it can be computed in $O(n^2 m^2)$ time.

## 5 Calibration and Decision-Making

Some of the most practically useful results of calibration are the guarantees provided for downstream decision-making [52]. Universal kernels guarantee distribution matching in principle, but in some cases require many samples. However, if we have information about the decision problem in advance, we can design calibration metrics that are only sensitive to errors relevant to decision-making. In this section, we show how to design such metrics and show that they enable us to (1) accurately evaluate the loss associated with downstream actions and (2) choose actions that are in a sense optimal.

We begin by introducing a general decision problem. For each example, a decision-maker chooses an action $a \in \mathcal{A}$ and incurs a loss $\ell(a, y)$ that depends on the outcome. Specifically, given information $z$ (such as the forecast), a decision policy $\delta : \mathcal{Z} \to \mathcal{A}$ assigns an action $\delta(z)$. The set of all such policies is denoted $\Delta(\mathcal{Z})$. The Bayes optimal policy under a label distribution $P_Y$ is denoted $\delta_\ell^*(P_Y) := \arg\min_{a \in \mathcal{A}} \mathbb{E}_P [\ell(a, Y)]$. Decision calibration [52] provides two important guarantees:

$$\mathbb{E}_Q [\ell(a, \widehat{Y}) \mid Z = z] = \mathbb{E}_P [\ell(a, Y) \mid Z = z], \quad \forall a \in \mathcal{A}, z \in \mathcal{Z} \qquad \text{(loss estimation)}$$
$$\mathbb{E}_P [\ell(\delta_\ell^*(Q_{Y|x}), Y) \mid Z = z] \leq \mathbb{E}_P [\ell(\delta(z), Y) \mid Z = z], \quad \forall z \in \mathcal{Z}, \delta \in \Delta(\mathcal{Z}) \qquad \text{(no regret)}$$

*Loss estimation* says that the expected loss of each action is equal under the true distribution and the forecast. This means we can estimate the loss on unlabeled data. *No regret* says that the Bayes optimal action according to the forecast $\delta_\ell^*(Q_{Y|x})$ achieves expected loss less than or equal to any decision policy $\delta \in \Delta(\mathcal{Z})$ that depends only on $z$. If we condition on the forecast by choosing $z = Q_{Y|x}$, then a decision-maker who only has access to the forecast is incentivized to act as if the forecast is correct. Additionally, we consider cases where the loss function is not known in advance. For example, when a vendor releases a foundation model to serve a large population of users, it is likely that different users have distinct loss functions. Thus, we also show settings in which we can achieve accurate loss estimation and no regret for all loss functions $\ell$ included in a family of losses $\mathcal{L}$.

**A General Recipe for Decision Calibration**  We now describe an approach to measure decision calibration using our kernel-based metrics. For each action $a \in \mathcal{A}$ and loss $\ell \in \mathcal{L}$, we want to measure the discrepancy between the expectation of the loss $\ell(a, Y)$ under $P$ and $Q$. To achieve this, we define the feature map that computes the losses of all actions $\phi(y) = (\ell(a, y))_{a \in \mathcal{A}, \ell \in \mathcal{L}}$. This assigns to $P$ and $Q$ the mean embeddings $\mu_P = \mathbb{E}_P [(\ell(a, Y))_{a \in \mathcal{A}, \ell \in \mathcal{L}}]$ and $\mu_Q = \mathbb{E}_Q [(\ell(a, \widehat{Y}))_{a \in \mathcal{A}, \ell \in \mathcal{L}}]$, respectively. Letting $\mathcal{F}$ be the unit ball of the Hilbert space associated with this feature map, the mean embedding form of MMD gives us that $\mathrm{MMD}(\mathcal{F}, P, Q)^2 = \|\mu_P - \mu_Q\|_2^2 = \sum_{a \in \mathcal{A}} \sum_{\ell \in \mathcal{L}} \left( \mathbb{E}_P [(\ell(a, Y))] - \mathbb{E}_Q [\ell(a, \widehat{Y})] \right)^2$. When either $\mathcal{A}$ or $\mathcal{L}$ is infinite, the corresponding sum is replaced by an integral. This metric measures whether each action has the same expected loss *marginally* under the true distribution and the forecasts. Thus, accurate loss estimation is only guaranteed marginally and the no regret result is relative to constant decision functions that choose the same action for all examples.

Next, we refine the decision calibration results to hold conditionally on a variable $Z$. Recall that the kernel over $y$ is given by $k_y(y, y') = \langle \phi(y), \phi(y') \rangle$. We lift this kernel into $\mathcal{Y} \times \mathcal{Z}$ by defining $k((y, z), (y', z')) = k_y(y, y') k_z(z, z')$ where $k_z$ is a universal kernel. The MMD metric corresponding to this kernel measures decision calibration conditioned on $Z$. If we choose $Z$ to be the forecast $Z = Q_{Y|X}$, then the no regret guarantee says that acting as if the forecasts were correct is an optimal decision policy among all policies depending only on the forecast.

Note that the feature maps defined above are infinite dimensional when $\mathcal{A}$ or $\mathcal{L}$ are infinite. However, applying the kernel trick sometimes allows us to estimate the MMD metric efficiently. Still, optimizing the MMD metric may be difficult in practice, especially when the conditioning variable $Z$ is very expressive. We suggest using the metric as a regularizer to incentivize decision calibration during training, and then still apply post-hoc recalibration if needed. We now give two concrete examples, one where the action space is infinite and one where the set of loss functions is infinite.

**Example 1** (Point Estimate Decision). Consider a regression problem in which a decision-maker chooses a point estimate $a \in \mathbb{R}$ and incurs the squared loss $\ell(a, y) = (a - y)^2$. The expected loss of an action $a$ is $\mathbb{E}_P [(a - Y)^2] = (a - \mathbb{E}_P [Y])^2 + \mathbb{E}_P [Y^2] - \mathbb{E}_P [Y]^2$. Observe that the expected loss only depends on $Y$ through the first two moments, $\mathbb{E}_P [Y]$ and $\mathbb{E}_P [Y^2]$. We use the representation $\phi(y) = (y, y^2)$, so that the kernel is $k(y, y') = \langle (y, y^2), (y', y'^2) \rangle$. From the mean embedding form of MMD, we see that $\mathrm{MMD}(\mathcal{F}, P, Q) = \|\mu_P - \mu_Q\|^2 = (\mathbb{E}[Y] - \mathbb{E}_Q[\widehat{Y}])^2 + (\mathbb{E}_P[Y^2] - \mathbb{E}_Q[\widehat{Y}^2])^2$. Therefore, the metric is zero exactly when the expected loss of each action $a$ is the same under the true distribution and the forecasts. The MMD metric incentivizes the forecasts to accurately reflect the marginal loss of each action. However, we often want to know the optimal action given a particular forecast, not marginally. We can now choose the conditioning variable to refine the guarantee. Let $Z$ be the forecast $Z = Q_{Y|X}$ and choose the kernel $k((y, z), (y', z')) = k_y(y, y') k_z(z, z')$ where $k_y(y, y') = \langle (y, y^2), (y', y'^2) \rangle$ as before and $k_z(z, z')$ is universal. For this kernel, the MMD will equal zero if and only if $\mathbb{E}_P [Y \mid Q_{Y|X}] = \mathbb{E}_Q [\widehat{Y} \mid Q_{Y|X}]$ and $\mathbb{E}_P [Y^2 \mid Q_{Y|X}] = \mathbb{E}_Q [\widehat{Y}^2 \mid$

$Q_{Y|X}]$ almost surely. The Bayes optimal action suggested by the forecast is optimal among decision policies that depend only on the forecast:

$$\mathbb{E}_P\left[\ell\left(\delta^*(Q_{Y|X}),Y\right)\right] \leq \mathbb{E}_P\left[\ell\left(\delta(Q_{Y|X}),Y\right)\right], \quad \forall \delta \in \Delta(Q_{Y|X}) \tag{5}$$

In other words, the decision maker is incentivized to behave as if the forecasts are correct.

**Example 2** (Threshold Decision). The above example had an infinite action space and a single loss function. Here, we consider a problem where the action space is small but the set of loss functions is infinite. This setting applies for a model vendor who serves a model to multiple decision makers, each with a potentially different loss function. For example, suppose that $Y \in [0,T]$ is a blood value for a patient for some $T > 0$, and the decision task is to determine whether $Y$ exceeds a threshold $t \in [0,T]$ that indicates eligibility for a clinical trial. Formally, the decision maker chooses a threshold $a \in \{\pm 1\}$ to minimize the loss $\ell_t(a,y) = \mathbb{1}\{a \neq \text{sign}(y-t)\}$. Our goal is to simultaneously calibrate our forecasts for all thresholds $t \in [0,T]$, i.e. $\mathcal{L} = \{\ell_t : t \in [0,T]\}$. In this case, we choose the feature map $\phi(y) = (\mathbb{1}\{y \geq t\})_{t \in [0,T]}$. The corresponding kernel is $k(y,y') = \langle \phi(y), \phi(y') \rangle = \min\{y,y'\}$. The mean embeddings are $\mathbb{E}_P[\phi(Y)] = P_Y$ and $\mathbb{E}_Q[\phi(\widehat{Y})] = Q_Y$, the marginal cdfs for $Y$ and $Y_Q$. Thus, the MMD metric measures $\|P_Y - Q_Y\|_2^2 = \int_{t=0}^{T}(P_Y(t) - Q_Y(t))^2\,dt$, which is zero exactly when $P_Y = Q_Y$. When $\text{MMD}(\mathcal{F},P,Q) = 0$, the expected loss of each action $a$ is the same under the true distribution and the forecasts, for all $\ell \in \mathcal{L}$. Similar to the previous example, we can achieve conditional decision calibration by adding a universal kernel over the conditioning variable.

**Discussion** The connections between decision-making and calibration have been previously studied by Zhao et al. [52] and Sahoo et al. [39]. Here, we extend results from Zhao et al. [52] to include infinite action spaces and further refine the decision guarantees to depend on an arbitrary conditioning variable. In Example 2, we connect our methods to the special case of threshold calibration [39]. Notably, our framework provides a strategy to optimize decision calibration at training time.

# 6 Experiments

The main goals of our experiments are to: (1) compare the performance our method with other trainable calibration metrics, and with standard post-hoc recalibration methods; (2) study the impact of tailoring the MMD kernel to a specific decision task; and (3) study whether trainable calibration metrics can be used to improve calibration across local neighborhoods of data.[2]

## 6.1 Datasets

We consider standard benchmark datasets and datasets relating to decision-making tasks. For each dataset, we randomly assign 70% of the dataset for training, 10% for validation, and 20% for testing.

**Regression Datasets** We use four tabular UCI datasets (SUPERCONDUCTIVITY [16], CRIME [34], BLOG [6], FB-COMMENT [41]), as well as the Medical Expenditure Panel Survey dataset (MEDICAL-EXPENDITURE [7]). They are common benchmarks in uncertainty quantification literature [23, 39]. The number of features ranges from $d = 53$ to $280$. The total number of examples $n$, and features $d$ for each dataset can be found in Table 3.

**Classification Datasets** We use five tabular UCI datasets: BREAST-CANCER [49], HEART-DISEASE [18], ONLINE-SHOPPERS [40], DRY-BEAN [1], and ADULT [2]. The datasets range from $m = 2$ to $7$ label classes, and $d = 16$ to $104$ features. The total number of examples $n$, features $d$, and classes $m$ for each dataset can be found in Table 4.

**Crop Yield Dataset** We introduce a CROP-YIELD dataset, where the task is to predict yearly wheat crop yield from weather data for different counties across the US. We use the NOAA database to track summary weather statistics across each year (temperature, precipitation, cooling/heating degree days), and the NASS database to track annual wheat crop yield. The dataset spans from 1990-2007, where data from 1995 was held-out for visualization, and the remaining data was used for training. We use this dataset to evaluate local calibration with respect to the latitude and longitude coordinates.

---

[2]Code to reproduce experiments can be found at https://github.com/kernel-calibration/kernel-calibration/.

Table 3: Comparison of model performance on five different regression datasets. Models were trained with two objectives: NLL and NLL + MMD (Ours). We display the metrics on the test set for each training procedure, both with and without post-hoc Quantile Calibration [23] fit to the validation set. $n$ is the number of examples in the dataset and $d$ is the number of features.

| Dataset | Training Objective | NLL ↓ | Quantile Cal. ↓ | Decision Cal. ↓ |
|---|---|---|---|---|
| CRIME $n = 1992$ $d = 102$ | NLL | -0.716 ± 0.007 | 0.220 ± 0.004 | 0.151 ± 0.009 |
| | + Post-hoc | -0.387 ± 0.012 | 0.154 ± 0.003 | 0.089 ± 0.005 |
| | NLL + MMD (*Ours*) | **-0.778 ± 0.008** | 0.164 ± 0.004 | 0.042 ± 0.011 |
| | + Post-hoc | -0.660 ± 0.014 | **0.105 ± 0.004** | **0.041 ± 0.005** |
| BLOG $n = 52397$ $d = 280$ | NLL | 0.997 ± 0.040 | 0.402 ± 0.006 | 4.087 ± 0.004 |
| | + Post-hoc | **0.624 ± 0.021** | 0.053 ± 0.001 | 4.090 ± 0.001 |
| | NLL + MMD (*Ours*) | 0.957 ± 0.008 | 0.302 ± 0.002 | **4.051 ± 0.002** |
| | + Post-hoc | 0.945 ± 0.007 | **0.042 ± 0.001** | 4.067 ± 0.001 |
| MEDICAL-EXPENDITURE $n = 33005$ $d = 107$ | NLL | 1.535 ± 0.000 | 0.078 ± 0.001 | 0.476 ± 0.002 |
| | + Post-hoc | 1.808 ± 0.002 | 0.059 ± 0.001 | 0.472 ± 0.002 |
| | NLL + MMD (*Ours*) | **1.532 ± 0.000** | 0.059 ± 0.001 | 0.438 ± 0.002 |
| | + Post-hoc | 1.780 ± 0.001 | **0.045 ± 0.000** | **0.431 ± 0.001** |
| SUPERCONDUCTIVITY $n = 21264$ $d = 81$ | NLL | 3.375 ± 0.008 | 0.062 ± 0.003 | 0.211 ± 0.003 |
| | + Post-hoc | 3.707 ± 0.007 | 0.066 ± 0.001 | 0.202 ± 0.003 |
| | NLL + MMD (*Ours*) | **3.269 ± 0.012** | **0.042 ± 0.003** | **0.182 ± 0.003** |
| | + Post-hoc | 3.643 ± 0.010 | 0.050 ± 0.002 | 0.186 ± 0.003 |
| FB-COMMENT $n = 40949$ $d = 53$ | NLL | 0.634 ± 0.016 | 0.334 ± 0.004 | 3.151 ± 0.003 |
| | + Post-hoc | 0.734 ± 0.015 | 0.063 ± 0.001 | 3.151 ± 0.002 |
| | NLL + MMD (*Ours*) | **0.605 ± 0.004** | 0.258 ± 0.003 | **3.131 ± 0.003** |
| | + Post-hoc | 0.639 ± 0.005 | **0.054 ± 0.001** | 3.138 ± 0.002 |

## 6.2 Experimental Setup and Baselines

**Models**   To represent the model uncertainty, we use a feed-forward neural network with three fully-connected layers. In the classification setting, the model outputs the logits for all $m$ classes. In the regression setting, the model outputs the predicted mean $\mu(Y)$ and variance $\sigma^2(Y)$, which are used to parameterize a Gaussian probability distribution. This allows us to tractably compute the inverse cdf $Q_{Y|X}^{-1}$ and perform differential sampling, enabling us to test our calibration metrics in multiple settings. See Appendix B for more details on model hyperparameter tuning.

**Training Objectives**   In the regression setting, we consider two training objectives: Negative Log-likelihood (NLL), and our trainable calibration metric as a regularizer (NLL + MMD). For MMD, we enforce a notion of individual calibration (Table 1) by choosing the conditioning variable $Z = X$ to match the distributions of $(X, \widehat{Y})$ and $(X, Y)$. More specifically, we optimize an MMD objective where the kernel $k$ is decomposed among the pair of random variables as $k((x, y), (x', y')) = k_X(x, x') \cdot k_Y(y, y')$. The form of $k_X$ and $k_Y$ varies across experiments. In the classification setting, we consider cross-entropy (XE) loss, and three trainable calibration metrics as regularizers: MMCE [26], KDE [37], and MMD (*Ours*).

**Baselines**   In the regression setting, we compare our approach against an uncalibrated forecaster trained with NLL, and with post-hoc recalibration using isotonic regression [23]. In the classification setting, we compare our approach against an uncalibrated forecaster trained with XE loss. Furthemore, we compare our approach to two trainable calibration metrics (MMCE [26] and KDE [37]). We compare all approaches against temperature-scaling [15], a standard post-hoc recalibration method. For both classification and regression, we train the post-hoc recalibration methods on a validation set, and subsequently evaluate on the held-out test set.

**Metrics**   To quantify predictive performance in regression, we evaluate the negative log-likelihood, quantile calibration error (QCE) [23], and decision calibration error (see Section 6.3). In classification, we compute accuracy, expected calibration error (calibration), and average Shannon entropy (sharpness) of the forecast. We also report kernel calibration error (KCE), defined as the MMD

Table 4: Experimental results for five tabular classification tasks, comparing MMCE regularization [26], KDE regularization [37], and MMD regularization (Ours), alongside a standard cross-entropy (XE) loss without post-hoc recalibration. Here, $n$ is the number of examples in the dataset, $d$ is the number of features, and $m$ is the number of classes.

| Dataset | Training Objective | Accuracy ↑ | ECE ↓ | Entropy ↓ | KCE ↓ |
|---|---|---|---|---|---|
| BREAST-CANCER | XE | 95.372 ± 0.160 | 0.194 ± 0.003 | 0.453 ± 0.004 | 6.781 ± 0.601 |
| $n = 569$ | XE + MMCE | 94.770 ± 0.147 | 0.060 ± 0.001 | 0.076 ± 0.002 | 0.392 ± 0.086 |
| $d = 30$ | XE + ECE KDE | 94.351 ± 0.163 | 0.062 ± 0.001 | 0.065 ± 0.001 | 0.225 ± 0.083 |
| $m = 2$ | XE + MMD (*Ours*) | **95.789 ± 0.060** | **0.052 ± 0.000** | **0.006 ± 0.000** | **0.014 ± 0.000** |
| HEART-DISEASE | XE | 55.904 ± 0.196 | 0.373 ± 0.003 | 1.512 ± 0.007 | 0.581 ± 0.016 |
| $n = 921$ | XE + MMCE | 60.787 ± 0.208 | 0.267 ± 0.002 | 0.947 ± 0.003 | 0.097 ± 0.002 |
| $d = 23$ | XE + ECE KDE | 50.036 ± 2.801 | 0.304 ± 0.012 | 1.392 ± 0.016 | 0.450 ± 0.050 |
| $m = 5$ | XE + MMD (*Ours*) | **61.516 ± 0.255** | **0.261 ± 0.003** | **0.945 ± 0.006** | **0.077 ± 0.002** |
| ONLINE-SHOPPERS | XE | 89.816 ± 0.037 | 0.129 ± 0.001 | 0.221 ± 0.003 | 0.022 ± 0.001 |
| $n = 12330$ | XE + MMCE | 89.933 ± 0.036 | 0.128 ± 0.001 | 0.220 ± 0.002 | -0.019 ± 0.001 |
| $d = 28$ | XE + ECE KDE | **90.019 ± 0.034** | **0.127 ± 0.000** | 0.225 ± 0.002 | -0.022 ± 0.001 |
| $m = 2$ | XE + MMD (*Ours*) | 89.976 ± 0.031 | **0.127 ± 0.001** | **0.218 ± 0.002** | **-0.026 ± 0.000** |
| DRY-BEAN | XE | 92.071 ± 0.025 | 0.113 ± 0.000 | 0.264 ± 0.001 | 0.061 ± 0.002 |
| $n = 13612$ | XE + MMCE | 92.772 ± 0.035 | 0.099 ± 0.000 | 0.224 ± 0.002 | 0.048 ± 0.004 |
| $d = 16$ | XE + ECE KDE | 92.760 ± 0.037 | 0.097 ± 0.000 | 0.232 ± 0.001 | 0.041 ± 0.003 |
| $m = 7$ | XE + MMD (*Ours*) | **92.894 ± 0.035** | **0.089 ± 0.000** | **0.174 ± 0.002** | **0.040 ± 0.004** |
| ADULT | XE | 84.528 ± 0.040 | 0.186 ± 0.000 | 0.320 ± 0.002 | -0.014 ± 0.002 |
| $n = 32561$ | XE + MMCE | 84.203 ± 0.042 | 0.191 ± 0.000 | 0.340 ± 0.002 | -0.015 ± 0.002 |
| $d = 104$ | XE + ECE KDE | 84.187 ± 0.045 | **0.178 ± 0.001** | 0.414 ± 0.002 | **-0.020 ± 0.003** |
| $m = 2$ | XE + MMD (*Ours*) | **84.565 ± 0.035** | **0.178 ± 0.001** | **0.315 ± 0.002** | -0.018 ± 0.001 |

objective for an RBF kernel over $\mathcal{X} \times \mathcal{Y}$. Each metric is computed on a held-out test set. We repeat all experiments across 50 random seeds and report the mean and standard error for each metric.

**Results** The results for regression and classification are shown in Table 3 and Table 4, respectively. In regression, our objective (NLL + MMD) achieves better QCE than the NLL objective across all datasets while maintaining comparable (or better) NLL scores on the test set. On most datasets, we find that our method also reduces the sharpness penalty induced by post-hoc calibration. Importantly, we also observe that post-hoc calibration and our method are complementary for calibrating forecasters; post-hoc methods can guarantee calibration on held-out data, and calibration regularization mitigates the sharpness penalty. For classification, we find that our method improves accuracy, ECE, and entropy relative to the baselines we tested for calibration regularization. These trends also generally hold with post-hoc calibration, where we observe that our method achieves better calibration and accuracy across most datasets (See Table 5).

**Computational Requirements** Relative to training with an unregularized NLL objective, our method with MMD regularization requires an average wall clock time per epoch of 1.2x for the 5 regression datasets, and 1.3x for the 5 classification datasets.

### 6.3 Experiment: Decision Calibration

One of the key advantages of our trainable calibration metrics is their flexibility in enforcing multiple notions of calibration. We consider a decision-making task to study the effectiveness of our method in improving decision calibration. The decision task is to choose an action $a \in \{\pm 1\}$, whose loss $\ell(a, y) = \mathbb{1}\{a \neq \text{sign}(y - c)\}$ depends only on whether the label $y$ exceeds a threshold $c \in \mathbb{R}$. Given a true distribution $P$ and a forecaster $Q_{Y|X}$, the **Decision Calibration Error** (DCE) is $\text{DCE}^2(P, Q) = \sum_{a \in \mathcal{A}} \|\mathbb{E}_P[\ell(Y, a)] - \mathbb{E}_X \mathbb{E}_{\widehat{Y} \sim Q_{Y|X}}[\ell(\widehat{Y}, a)]\|_2^2$.

**MMD Objective** To optimize for decision calibration, we tailor the kernels $k_X, k_Y$ to reflect the decision problem. More concretely, we choose a kernel function that penalizes predicted labels which are on the wrong side of $c$, namely $k_Y(y, y') = +1$ if $\text{sign}(y - c) = \text{sign}(y' - c)$, and $-1$ otherwise.

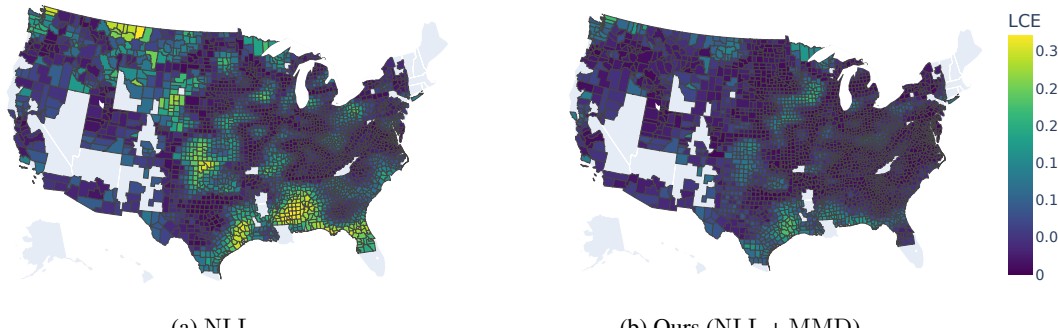

|          |          |
|:--------:|:--------:|
| (a) NLL  | (b) Ours (NLL + MMD) |

Figure 1: We visualize Local Calibration Error for a forecaster trained to predict annual wheat crop-yield based on climate and geospatial data. The standard NLL objective (**Left**) leads to a forecaster that is miscalibrated across local geospatial neighbourhoods, as seen by areas of brighter color. Our kernel-based calibration objective (**Right**) leads to better calibration across local neighborhoods.

To allow for gradient based optimization, we relax the objective to yield the differentiable kernel $k_Y(y, y') = \tanh(y - c) \cdot \tanh(y' - c)$. Notice that $k_Y(y, y') \approx +1$ when $\text{sign}(y - c) = \text{sign}(y - c)$, and $-1$ otherwise. To encourage decision calibration for all features, we let $k_X$ be an RBF kernel.

**Results**   The results in Table 3 demonstrate that the MMD training objective tailored to the decision problem achieves the best decision calibration across all datasets. Additional results in Appendix B.3 show that using an MMD kernel tailored to a decision problem (i.e. the $\tanh$ kernel) improves decision calibration scores across all datasets.

### 6.4   Experiment: Local Calibration

**Setup**   To provide intuition on how our metrics can facilitate calibration in local neighbourhoods of features, we study how location affects the reliability of forecasts on the CROP-YIELD regression dataset. We tailor the kernels $k_X, k_Y$ to local calibration for geospatial neighborhoods. Namely, we select $\phi(x)$ which extracts the geospatial features (latitude, longitude) from the full input vector $x$. We then define $k_X(x, x') = k_{\text{RBF}}(\phi(x), \phi(x'))$ and $k_Y(y, y') = k_{\text{RBF}}(y, y')$.

**Metric**   Extending prior work on local calibration [27], given a cdf forecaster $Q_{Y|X}$, a feature embedding $\phi(\cdot)$ and kernel $k$, we define **Local Calibration Error** (LCE) for regression is $\text{LCE}(x; c) = \left( \sum_i \mathbb{1}\{y_i \leq Q_{Y|x_i}^{-1}(c)\} \cdot k(\phi(x), \phi(x_i)) \right) / \left( \sum_i k(\phi(x), \phi(x_i)) \right) - c$. The total LCE is computed as $\text{LCE}_{\text{total}}(x) = \frac{1}{B} \sum_{i=1}^{B} \text{LCE}(x; c_i)^2$, where $c_i = \frac{i}{B}$. Intuitively, LCE measures the QCE across local regions determined by $\phi(\cdot)$. We visualize the results in Figure 1 for $B = 20$.

## 7   Conclusion

We introduced a flexible class of trainable calibration metrics based on distribution matching using MMD. These methods can be effectively combined with existing post-hoc calibration methods to achieve strong calibration guarantees while preserving sharpness.

**Limitations**   Optimizing the MMD metric requires that samples from the forecast are differentiable with respect to the model parameters. This restricts the family of applicable forecasters, but differentiable relaxations such as Gumbel-softmax can mitigate this challenge. Furthermore, in practice it is often not possible to achieve zero calibration error via regularization due to computational and data limitations. To mitigate this limitation, we suggest pairing our calibration regularizers with post-hoc calibration methods. Lastly, the scale of the MMD metric is sensitive to the chosen kernel (e.g., the bandwidth of a RBF kernel), so it can be difficult to understand how calibrated a model is in an absolute sense based on the MMD metric.

## 8 Acknowledgements

CM is supported by the NSF GRFP. This research was supported in part by NSF (#1651565), ARO (W911NF-21-1-0125), ONR (N00014-23-1-2159), and CZ Biohub.

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

# A    Expressing Popular Forms of Calibration as Distribution Matching

## A.1    Calibration in Regression

Here, we show how the forms of calibration listed in Table 1 can be expressed in terms of conditional distribution matching. Note that many of these forms of calibration are sensible in the classification setting as well (e.g., marginal, group, and decision calibration).

**Quantile Calibration**    Let $Y$ be a continuous random variable. A forecaster is quantile calibrated if

$$\mathbb{P}\left(Q_{Y|X}(Y) \leq t\right) = t, \quad \forall t \in [0, 1]. \tag{6}$$

This is equivalent to stating that $Q_{Y|X}(Y)$ is uniform on the interval $[0, 1]$. For continuous random variables, the probability integral transform $P_{Y|X}(Y)$ is uniform on the interval $[0, 1]$. Thus, quantile calibration can be written as $Q_{Y|X}(Y) \triangleq P_{Y|X}(Y)$, where $\triangleq$ denotes equality in distribution.

**Threshold Calibration**    Let $Y$ be a continuous random variable. A forecaster satisfies threshold calibration [39] if $\mathbb{P}(Q_{Y|X}(Y) \leq t \mid Q_{Y|X}(y) \leq c) = t$ for all $t \in [0, 1]$, $c \in [0, 1]$, and $y \in \mathcal{Y}$. This states that $Q_{Y|X}(Y)$ is uniform, conditioned on the event $Q_{Y|X}(y) \leq c$. Noting that $P_{Y|X}(Y)$ is uniform, the constraint is then equivalent to the following:

$$Q_{Y|X}(Y) \triangleq P_{Y|X}(Y) \mid \mathbb{1}\left\{Q_{Y|X}(y) \leq c\right\}, \quad \forall c \in [0, 1], y \in \mathcal{Y} \tag{7}$$

**Marginal Calibration**    A forecaster is marginally calibrated [11] if $\mathbb{E}[Q_{Y|X}(y)] = P_Y(y)$ for all $y \in \mathcal{Y}$. Recall that the distribution of $\widehat{Y}$ is defined by $(\widehat{Y} \mid X = x) \sim Q_{Y|x}$. Thus, the marginal distribution of $\widehat{Y}$ is given by $\widehat{Y} \sim \mathbb{E}[Q_{Y|X}]$. So marginal calibration states that $\widehat{Y}$ and $Y$ have the same marginal cdf or, equivalently, $\widehat{Y} \triangleq Y$.

**Decision Calibration**    Zhao et al. [52] introduce the concept of $\mathcal{L}^K$ decision calibration for multiclass classification, where $\mathcal{Y} = \{0, 1\}^C$ is the set of onehot vectors over $C$ classes.

**Definition A.1** ($\mathcal{L}^K$-Decision Calibration, Zhao et al. [52])**.** Let $\mathcal{L}^K$ be the set of all loss functions with $K$ actions $\mathcal{L}^K = \{\ell : \mathcal{Y} \times \mathcal{A} \to \mathbb{R}, |\mathcal{A}| = K\}$, we say that a prediction $Q_{Y|X}$ is $\mathcal{L}^K$-decision calibrated (with respect to $P_{Y|X}$) if $\forall \ell \in \mathcal{L}^K$ and $\delta \in \Delta_{\mathcal{L}^K}$

$$\mathbb{E}_X \mathbb{E}_{\widehat{Y} \sim Q_{Y|X}}[\ell(\widehat{Y}, \delta(Q_{Y|X}))] = \mathbb{E}_X \mathbb{E}_{Y \sim P_{Y|X}}[\ell(Y, \delta(Q_{Y|X}))] \tag{8}$$

where $\Delta_{\mathcal{L}^K}$ is the set of all Bayes decision rules over loss functions with $K$ actions.

They show that $\mathcal{L}^K$ decision calibration can equivalently be expressed as requiring that $\mathbb{E}[(\widehat{Y} - Y)\mathbb{1}\{\delta(Q_{Y|X}) = a\}] = 0$ for all $\delta \in \Delta_{\mathcal{L}^K}$. We can rewrite this as

$$\mathbb{E}[\widehat{Y}|\delta(Q_{Y|X}) = a] = \mathbb{E}[Y|\delta(Q_{Y|X}) = a], \quad \forall \delta \in \Delta_{\mathcal{L}^K}, a \in \mathcal{A} \tag{9}$$

note that $\mathbb{E}[Y|\delta(Q_{Y|X}) = a]$ gives the conditional distribution of $(Y \mid \delta(Q_{Y|X}) = a)$ due to the onehot representation of $Y$. Thus, it is equivalent to require that

$$\left(\widehat{Y} \triangleq Y\right) \mid \delta(Q_{Y|X}), \quad \forall \delta \in \Delta_{\mathcal{L}^K} \tag{10}$$

where $\triangleq$ denotes equality in distribution.

**Group Calibration**    Group calibration [20] says that forecasts are marginally calibrated for each subgroup (e.g., subgroups defined by gender or age). We can write this as $\widehat{Y} \triangleq Y \mid \text{Group}(X)$, where $\text{Group}(X)$ is a variable indicating which subgroup includes the feature vector $X$.

**Distribution Calibration**    A forecaster satisfies distribution calibration [42] if for all possible forecasts $Q_0$ and labels $y \in \mathcal{Y}$, we have $\mathbb{P}\left(Y \leq y \mid Q_{Y|X} = Q_0\right) = Q_0(y)$. This is equivalent to requiring $P_{Y|X}(y) \mid Q_{Y|X} = Q_{Y|X}(y)$ for all $y \in \mathcal{Y}$. Since the cdfs take the same value at all $y \in \mathcal{Y}$, the random variables $Y$ and $\widehat{Y}$ have the same distribution, giving us that $\widehat{Y} \triangleq Y \mid Q_{Y|X}$.

**Individual Calibration**   Let $Y$ be a real-valued continuous random variable. Individual calibration [51] states that $\mathbb{P}\left(Q_{Y|x}(Y) \leq t\right) = t$, for all $t \in [0, 1]$ and $x \in \mathcal{X}$. Letting $U \sim \mathrm{Unif}(0, 1)$, we can write this as

$$Q_{Y|X}(Y) \triangleq P_{Y|X}(Y) \mid X \tag{11}$$

$$Q_{Y|X}^{-1}(Q_{Y|X}(Y)) \triangleq Q_{Y|X}^{-1}(P_{Y|X}(Y)) \mid X \tag{12}$$

$$Y \triangleq Q_{Y|X}^{-1}(U) \mid X \tag{13}$$

$$Y \triangleq \widehat{Y} \mid X \tag{14}$$

**Local Calibration**   A forecaster is locally calibrated [27] if $Q_{Y|X}$ and $P_{Y|X}$ are equal, on average, in any neighborhood of the feature space. Similarity in feature space is defined by a kernel $k(x, x')$. In regression, local calibration for quantiles says that

$$\frac{\mathbb{E}[\mathbb{1}\left\{Q_{Y|X}(Y) \leq t\right\} k(X, x)]}{\mathbb{E}[k(X, x)]} = t, \quad \forall t \in [0, 1], x \in \mathcal{X}. \tag{15}$$

This can equivalently be stated as

$$\mathbb{E}[\mathbb{1}\left\{Q_{Y|X}(Y) \leq t\right\} k(X, x)] = \mathbb{E}[\mathbb{1}\left\{P_{Y|X}(Y) \leq t\right\} k(X, x)], \quad \forall t \in [0, 1], x \in \mathcal{X} \tag{16}$$

or, in terms of the canonical feature mapping,

$$\mathbb{E}[\mathbb{1}\left\{Q_{Y|X}(Y) \leq t\right\} \langle \phi(X), \phi(x) \rangle] = \mathbb{E}[\mathbb{1}\left\{P_{Y|X}(Y) \leq t\right\} \langle \phi(X), \phi(x) \rangle], \quad \forall t \in [0, 1], x \in \mathcal{X}. \tag{17}$$

Thus, it is sufficient for $\mathbb{E}[Q_{Y|X}]$ and $\mathbb{E}[P_{Y|X}]$ to have the same distribution conditioned on the feature mapping $\phi(X)$. This can be written succinctly as

$$Y \triangleq \widehat{Y} \mid \phi(X) \tag{18}$$

## A.2   Calibration in Classification

Here, we express three popular forms of calibration from the classification literature—namely, canonical, top-label, and marginal calibration [44, Eq (1-3)]—in terms of distribution matching constraints. For the entirety of this section, $Y$ is a discrete random variable taking values in the set $\mathcal{Y} = \{1, 2, \ldots, m\}$, and $q_{Y|x}$ is the probability mass function of the forecast given the features $X = x$.

**Canonical Calibration**   A forecaster satisfies canonical calibration if $\mathbb{P}\left(Y = y \mid q_{Y|X}\right) = q_{Y|X}(y)$ for all $y \in \mathcal{Y}$. This says that the label follows the same distribution under the forecast and the true distribution, conditioned on $q_{Y|X}$. In other words, $Y \triangleq \widehat{Y} \mid q_{Y|X}$.

**Top-Label Calibration**   A forecaster satisfies top-label calibration if $\mathbb{P}\left(Y = Y^* \mid q_{Y|X}(Y^*)\right) = q_{Y|X}(Y^*)$, where $Y^* := \arg\max_{y \in \mathcal{Y}} q_{Y|X}(y)$ is the mode of the forecast. This equation can be rewritten in terms of the indicators

$$\mathbb{E}\left[\mathbb{1}\left\{Y = Y^*\right\} \mid q_{Y|X}(Y^*)\right] = \mathbb{E}\left[\mathbb{1}\{\widehat{Y} = Y^*\} \mid q_{Y|X}(Y^*)\right]$$

and the indicator variables are equal in expectation if and only if they are equal in distribution. Thus, top-label calibration is equivalent to $\mathbb{1}\left\{Y = Y^*\right\} \triangleq \mathbb{1}\{\widehat{Y} = Y^*\} \mid q_{Y|X}(Y^*)$.

**Marginal Calibration**   A forecaster satisfies marginal calibration if $\mathbb{P}\left(Y = y \mid q_{Y|X}(y)\right) = q_{Y|X}(y)$ for all $y \in \mathcal{Y}$. Similar to top-label calibration, this can be written as

$$\mathbb{E}\left[\mathbb{1}\left\{Y = y\right\} \mid q_{Y|X}(y)\right] = \mathbb{E}\left[\mathbb{1}\{\widehat{Y} = y\} \mid q_{Y|X}(y)\right]$$

and the indicator variables are equal in expectation if and only if they are equal in distribution. Thus, marginal calibration can be written as $\mathbb{1}\left\{Y = y\right\} \triangleq \mathbb{1}\{\widehat{Y} = y\} \mid q_{Y|X}(y)$, for all $y \in \mathcal{Y}$.

## B   Experimental Setup and Additional Results

### B.1   Reproducibility

In order to reproduce the experiments and results obtained in this paper, we provide details about the hyperparameter tuning and required computational resources.

**Hyperparameter Search:**   We use TPES [3] to perform hyperparameter optimization. For all experiments, we vary:

- Layer sizes between 32 and 512
- RBF kernel bandwidths between 0.001 and 200
- Batch sizes between 16 and 512, with and without batch normalization
- Learning rates between $10^{-7}$ and $10^{-1}$.
- The loss mixture weight $\lambda$ (as in $\text{NLL} + \lambda \cdot \text{MMD}$ and $\text{XE} + \lambda \cdot \text{MMD}$) between 0.1 and 1000.

For each dataset, we test 100 hyperparameter settings for each training objective for regression $\{\text{NLL}, \text{NLL} + \text{MMD}\}$, and classification setting $\{\text{XE}, \text{XE} + \text{MMCE}, \text{XE} + \text{KDE}, \text{XE} + \text{MMD}\}$. For each run, we enable early-stopping if the validation loss does not improve for more than 50 epochs. In the regression setting, we pick the best performing run based on the validation NLL. For classification, we select the best performing run based on the validation accuracy, with validation ECE used for break ties. For each model and dataset, the best performing model is then re-run with 50 random seeds to gather information about standard errors and statistical significance. To capture variability across different seeds, we report the standard error of the mean in Tables 3, 4, and 5.

**Kernel Bandwidth**   We select the RBF kernel bandwidth for training on each dataset using the aforementioned hyperparameter optimization. For each dataset in Tables 4 and 5, the KDE metric we report is for the bandwidth selected through a held-out validation set, scaled so that the error is of a comparable order of magnitude across datasets (since multiplying a kernel by a positive scalar preserves the kernel properties).

**Baselines**   In the classification setting, we compare our approach to two state-of-the-art trainable calibration methods, MMCE [26] and KDE [37]. For MMCE, we follow the same experimental setup as the official author implementation, where the loss mixture weight $\lambda$ and the Laplacian kernel bandwidth are chosen via a held-out validation set. For KDE, we use the $L_1$ canonical calibration using the $ECE^{KDE}$ estimator, calculated according to Equation (9) from Popordanoska et al. [37]. We follow the same experimental setup as the official author implementation, where the kernel bandwidth is chosen using leave-one-out-likelihood (LOO MLE), and the loss mixture weight $\lambda$ is chosen via a held-out validation set.

**Computational Requirements:**   All experiments were conducted on a single CPU machine (Intel(R) Xeon(R) Gold 6342 CPU @ 2.80GHz), utilizing 8 cores per experiment. When run on a CPU, the training time for a single model ranges from 5 to 15 minutes, depending on the dataset size. It is possible to perform a full hyperparameter sweep using this setup in ∼24 hours.

To accelerate training, we have also run some experiments using an 11GB NVIDIA GeForce GTX 1080 Ti. When run on a GPU, the training time for a single model ranges from 2 to 5 minutes, depending on the dataset size, and a hyperparameter sweep takes ∼10 hours.

**US Agriculture Dataset:**   We introduce a new CROP-YIELD dataset for predicting yearly wheat crop yield from geospatial weather data for different counties across the US. We use NOAA database which contains weather data from thousands of stations across the United States. For each county, we track the weather sequence of each year into a few summary statistics for each month (average/maximum/minimum temperatures, precipitation, cooling/heating degree days). To study how climate change and location affect crop yield, we match crop yield data from the NASS dataset to weather stations in the NOAA database, and learn a function to predict crop yield from weather data. We specifically considered the crop yield of wheat across different counties in the US between 1990-2007. Since crop yield data for 1995 is the least sparse, we hold out data from 1995 for visualization, and train our forecasters on the remaining data.

## B.2 Additional Experimental Results

Table 5: Performance on classification tasks, comparing MMCE regularization [26], KDE regularization [37], and MMD regularization (*Ours*), alongside a standard cross-entropy (XE) loss. Temperature scaling is used for post-hoc recalibration of all methods. $n$ is the number of examples in the dataset, $d$ is the number of features, and $m$ is the number of classes.

| Dataset | Training Objective with Temp. Scaling | Accuracy ↑ | ECE ↓ | Entropy ↓ | KCE ↓ |
|---|---|---|---|---|---|
| BREAST-CANCER | XE | 95.372 ± 0.160 | 0.105 ± 0.031 | 0.056 ± 0.007 | 0.028 ± 0.005 |
| $n = 569$ | XE + MMCE | 94.770 ± 0.147 | **0.052 ± 0.001** | 0.011 ± 0.001 | **0.019 ± 0.000** |
| $d = 30$ | XE + ECE KDE | 94.351 ± 0.163 | 0.074 ± 0.018 | 0.010 ± 0.001 | 0.022 ± 0.003 |
| $m = 2$ | XE + MMD (*Ours*) | **95.789 ± 0.060** | **0.052 ± 0.000** | **0.004 ± 0.000** | **0.019 ± 0.000** |
| HEART-DISEASE | XE | 55.904 ± 0.196 | 0.325 ± 0.003 | 1.116 ± 0.008 | 0.077 ± 0.000 |
| $n = 921$ | XE + MMCE | 60.787 ± 0.208 | 0.262 ± 0.002 | 0.927 ± 0.004 | 0.067 ± 0.000 |
| $d = 23$ | XE + ECE KDE | 50.036 ± 2.801 | 0.276 ± 0.011 | 1.192 ± 0.033 | 0.082 ± 0.002 |
| $m = 5$ | XE + MMD (*Ours*) | **61.516 ± 0.255** | **0.252 ± 0.003** | **0.860 ± 0.003** | **0.066 ± 0.000** |
| ONLINE-SHOPPERS | XE | 89.816 ± 0.037 | 0.130 ± 0.000 | 0.227 ± 0.001 | **0.050 ± 0.000** |
| $n = 12330$ | XE + MMCE | 89.933 ± 0.036 | 0.128 ± 0.000 | 0.220 ± 0.000 | 0.051 ± 0.000 |
| $d = 28$ | XE + ECE KDE | **90.019 ± 0.034** | 0.125 ± 0.000 | 0.216 ± 0.001 | **0.050 ± 0.000** |
| $m = 2$ | XE + MMD (*Ours*) | 89.976 ± 0.031 | **0.126 ± 0.000** | **0.214 ± 0.001** | **0.050 ± 0.000** |
| DRY-BEAN | XE | 92.071 ± 0.025 | 0.102 ± 0.000 | 0.206 ± 0.001 | **0.011 ± 0.000** |
| $n = 13612$ | XE + MMCE | 92.772 ± 0.035 | 0.092 ± 0.000 | 0.188 ± 0.001 | **0.011 ± 0.000** |
| $d = 16$ | XE + ECE KDE | 92.760 ± 0.037 | 0.091 ± 0.000 | 0.190 ± 0.000 | **0.011 ± 0.000** |
| $m = 7$ | XE + MMD (*Ours*) | **92.894 ± 0.035** | **0.086 ± 0.000** | **0.187 ± 0.001** | **0.011 ± 0.000** |
| ADULT | XE | 84.528 ± 0.040 | 0.188 ± 0.000 | **0.330 ± 0.000** | 0.022 ± 0.000 |
| $n = 32561$ | XE + MMCE | 84.203 ± 0.042 | 0.190 ± 0.000 | 0.335 ± 0.000 | 0.022 ± 0.000 |
| $d = 104$ | XE + ECE KDE | 84.187 ± 0.045 | 0.177 ± 0.001 | 0.338 ± 0.001 | 0.023 ± 0.000 |
| $m = 2$ | XE + MMD (*Ours*) | **84.565 ± 0.035** | **0.174 ± 0.000** | 0.334 ± 0.000 | **0.020 ± 0.000** |

Table 6: Performance versus the number of simulated samples per forecast used for the plug-in MMD estimate in regression tasks. All other hyperparameters are held constant, including the number of training steps. Increasing the number of samples gives a better estimate of the MMD objective, generally leading to better performance at the cost of additional compute.

| # Samples | CRIME | | | BLOG | | | MEDICAL-EXPENDITURE | | |
|---|---|---|---|---|---|---|---|---|---|
| | NLL | QCE | DCE | NLL | QCE | DCE | NLL | QCE | DCE |
| 1 | -0.703 | 0.198 | 0.088 | 0.952 | 0.379 | 4.171 | 1.546 | 0.071 | 0.448 |
| 2 | -0.755 | 0.188 | 0.133 | 0.959 | 0.444 | 4.228 | 1.535 | 0.07 | 0.427 |
| 5 | -0.72 | 0.154 | 0.043 | 0.96 | 0.395 | 4.09 | 1.539 | 0.071 | **0.419** |
| 10 | -0.777 | 0.154 | 0.06 | 0.85 | 0.416 | 4.077 | 1.54 | 0.072 | 0.459 |
| 20 | -0.772 | 0.157 | 0.054 | 0.835 | 0.415 | 4.031 | 1.538 | **0.063** | 0.431 |
| 50 | -0.779 | **0.150** | **0.041** | 0.847 | 0.432 | 4.016 | **1.531** | 0.065 | 0.447 |
| 100 | -0.781 | 0.153 | 0.043 | **0.829** | 0.386 | **3.978** | 1.536 | 0.064 | 0.449 |
| 200 | **-0.782** | 0.151 | 0.043 | **0.829** | 0.373 | **3.978** | 1.535 | 0.064 | 0.442 |

| # Samples | SUPERCONDUCTIVITY | | | FB-COMMENT | | |
|---|---|---|---|---|---|---|
| | NLL | QCE | DCE | NLL | QCE | DCE |
| 1 | 3.369 | 0.053 | 0.182 | 0.637 | 0.268 | 3.15 |
| 2 | 3.35 | 0.062 | 0.197 | 0.477 | 0.294 | 3.14 |
| 5 | 3.311 | 0.038 | 0.216 | **0.409** | 0.276 | 3.146 |
| 10 | 3.333 | 0.036 | 0.208 | 0.59 | 0.278 | **3.102** |
| 20 | 3.298 | 0.055 | 0.262 | 0.569 | 0.258 | 3.119 |
| 50 | 3.345 | 0.041 | 0.208 | 0.479 | **0.223** | 3.14 |
| 100 | 3.318 | 0.036 | **0.181** | 0.469 | 0.23 | 3.149 |
| 200 | **3.291** | **0.034** | 0.182 | 0.458 | 0.231 | 3.137 |

## B.3 Experiment: Decision Calibration

As an illustrative example of how trainable calibration metrics can improve calibration for a specific decision-making problem, we consider a common real-world decision problem, requiring us to make an action whose utility depends only on whether the label $y$ is greater or less than a particular threshold $c$. Concretely, the decision loss $\ell : \mathcal{A} \times \mathcal{Y} \to \mathbb{R}$, for an action $a \in \{-1, +1\}$ and a label $y \in \mathcal{Y}$, is defined as $\ell(a, y) = \mathbb{1}\{a \neq \mathrm{sign}(y - c)\}$.

**Setup** In order to achieve decision calibration, we tailor our kernels $k_X, k_Y$ to closely follow the decision problem. More concretely, we choose a kernel function that penalizes predicted labels which are on the wrong side of $c$, namely $k_Y(y, y') = \tanh(y - c) \cdot \tanh(y' - c)$. Notice that $k_Y(y, y') \approx +1$ when $\mathrm{sign}(y - c) = \mathrm{sign}(y - c)$, and $-1$ otherwise. Thus, the calibration metric gives the model training feedback that is well aligned with the decision problem. To encourage decision calibration across all features, we set $k_X$ to be the universal RBF kernel.

As part of this experiment, we investigate whether tailoring an MMD kernel to the decision problem will improve decision calibration over using a universal RBF kernel for both $k_X, k_Y$.

**Metric** To evaluate the effectiveness of methods, we compute the standard Decision Calibration Error (DCE) across all regression datasets, defined as

$$\mathrm{DCE}^2 = \sum_{a \in \mathcal{A}} \left\lVert \mathbb{E}_P[\ell(Y, a)] - \mathbb{E}_X \mathbb{E}_{\widehat{Y} \sim Q_{Y|X}}[\ell(\widehat{Y}, a)] \right\rVert_2^2, \tag{19}$$

where $P$ is the true distribution over $\mathcal{X} \times \mathcal{Y}$, and $Q_{Y|X}$ is a forecaster.

**Additional Results: Choice of kernel** We present two sets of results. The first set of results is shown in Table 7, comparing models trained using our objective (NLL + MMD), where one uses a kernel $k_Y$ that is tailored to the decision problem ($\tanh$), while the other uses a universal RBF kernel. As described in Section 6.3, we observe that using the $\tanh$ kernel clearly achieves better decision calibration across all dataset, with minimal degradation in sharpness, i.e. NLL. This demonstrates that we can improve decision calibration using our trainable calibration metrics by tailoring our kernels to closely follow the decision problem.

The second set of results, shown in Figure 2, further demonstrates the relationship between our trainable calibration metrics and decision calibration throughout model training. By tailoring our kernels to closely follow the decision problem, we observe that decision calibration is optimized throughout training. In contrast, using objectives that are agnostic to the decision problem (i.e. NLL, or MMD with RBF kernels) will lead to noticeably worse decision calibration, with little to no improvement throughout training. The same trend is observed on a held-out validation set, as shown in Figure 3.

| Dataset | Kernel ($k_Y$) | NLL ↓ | Decision Cal. ↓ |
|---|---|---|---|
| CRIME | RBF | **-0.778 ± 0.008** | 0.042 ± 0.011 |
| ($n = 1992, d = 102$) | tanh | -0.657 ± 0.010 | **0.027 ± 0.006** |
| BLOG | RBF | 0.957 ± 0.008 | 4.051 ± 0.002 |
| ($n = 52397, d = 280$) | tanh | **0.934 ± 0.007** | **3.052 ± 0.001** |
| MEDICAL-EXPENDITURE | RBF | **1.530 ± 0.000** | 0.447 ± 0.002 |
| ($n = 33005, d = 107$) | tanh | 1.532 ± 0.000 | **0.438 ± 0.002** |
| SUPERCONDUCTIVITY | RBF | **3.269 ± 0.012** | 0.182 ± 0.003 |
| ($n = 21264, d = 81$) | tanh | 3.311 ± 0.009 | **0.127 ± 0.003** |
| FB-COMMENT | RBF | 0.605 ± 0.004 | 3.131 ± 0.003 |
| ($n = 40949, d = 53$) | tanh | **0.598 ± 0.003** | **2.342 ± 0.003** |

Table 7: Comparison of model performance trained with our training objective (NLL + MMD), using different kernels for $k_Y$ in the MMD kernel. The RBF kernel is a commonly-used universal kernel, while the $\tanh$ kernel is specifically designed to match the decision task for decision calibration. All metrics are reported on the test set.

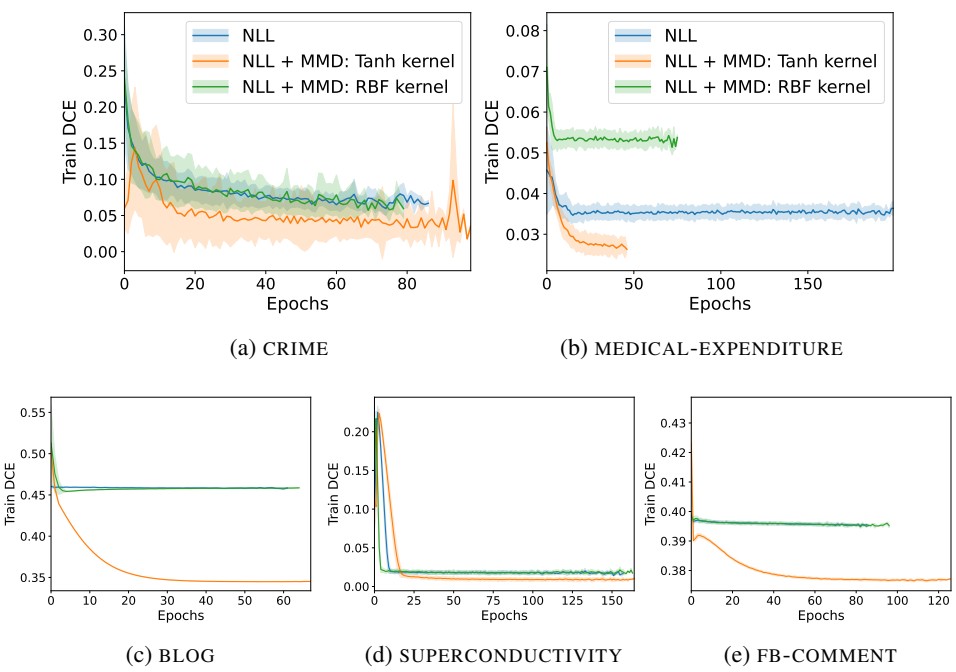

(a) CRIME

(b) MEDICAL-EXPENDITURE

(c) BLOG

(d) SUPERCONDUCTIVITY

(e) FB-COMMENT

Figure 2: We visualize the Decision Calibration Error (DCE) evaluated throughout training on the training data, for a Gaussian forecaster trained using different objectives. Our method, by tailoring the kernels to a specific decision problem, achieves the best DCE among baselines across all datasets. Using our method, we observe continuous improvement in DCE throughout training. Error bars denote the standard deviation computed over 50 random trials.

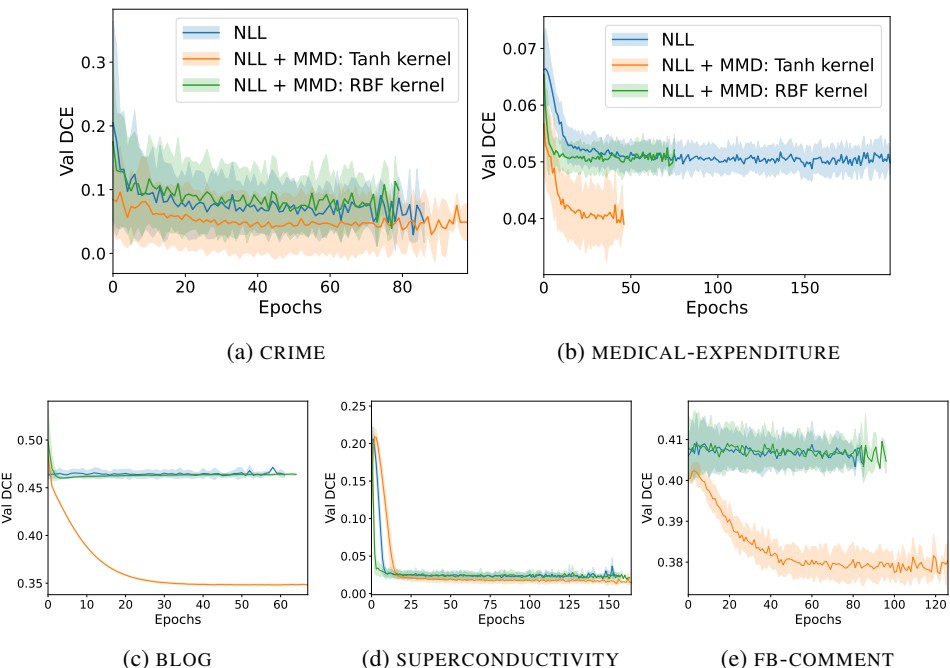

(a) CRIME

(b) MEDICAL-EXPENDITURE

(c) BLOG

(d) SUPERCONDUCTIVITY

(e) FB-COMMENT

Figure 3: We visualize the Decision Calibration Error (DCE) evaluated throughout training on held-out validation data. Our method, by tailoring the kernels to a specific decision problem, achieves the best DCE among baselines across all datasets. Error bars denote the standard deviation computed over 50 random trials.

