# OpenReview forum: "Calibration by Distribution Matching: Trainable Kernel Calibration Metrics"
_NeurIPS.cc/2023/Conference — NeurIPS 2023 poster_

### Official Review · Reviewer_UW9V · 2023-06-24

**Soundness:** 3 good
**Presentation:** 3 good
**Contribution:** 3 good
**Rating:** 6
**Confidence:** 3

**Summary:**

Accurate probability prediction is a crucial aspect of a trustworthy model, especially for deep neural networks. This paper focuses on the model calibration problem and proposes a unified framework via distribution matching. To leverage the distribution matching, they used
Maximum Mean Discrepancy to estimate the miscalibration, where MMD is a popular distribution measure and used in many distribution alignment problems in recent years. The main usage of MMD in this paper is to treat it as a trainable regularizer. In the experiment section,
they demonstrate their effectiveness in performing proper distribution alignments.

**Strengths:**

1. They introduced a unified framework for calibration metrics as distribution-matching problems, which is a very interesting view for uncertainty calibration.
2. They proposed a training-time model calibration method by introducing kernel-based calibration metrics.

**Weaknesses:**

The paper does not have main weakness, if some, they are:

1. How to apply this method to multi-class classification? I did not see the related experiments. It seems that all the used datasets are regression datasets, but Line 6 said it is also suitable for classification tasks.

2. Deep comparisons with similar kernel methods, like [1]. I did not see the description of their difference. They also used the kernel method to perform model calibration.

[1] Trainable Calibration Measures For Neural Networks From Kernel Mean Embeddings. ICML 2018

3. What are the n and d in Table 2? Why do you use the different n and d regarding different datasets?

4. How to demonstrate the prediction performance improvement using the proposed methods? I did not find the clue in Table 2.

**Questions:**

Please refer to the weaknesses.

**Limitations:**

Yes

---

> ### Author Rebuttal · Authors · 2023-08-10
>
> Thank you for your time and feedback! We have focused on improving the related works section to include more direct comparisons to [Kumar et al., 2018] and other kernel-based calibration metrics. We have included updated comparisons in our general author response. We respond to your questions and comments below:
>
> > **How to apply this method to multi-class classification?**
>
> See Table 3 in the PDF attached to this response for a description of how three popular forms of calibration in classification fit into our framework. In new experiments, we implement the most stringent of the three forms (labeled “canonical” in Table 3 of the PDF). We find that regularizing with our proposed approach achieves better ECE, entropy, and accuracy than when training with the MMCE regularization of [Kumar et al, 2018], or without regularization. We find that these results hold both with and without post-hoc temperature scaling (see Tables 1 and 2 of the PDF attached to this response).
>
> > **Deep comparisons with similar kernel methods, like [Kumar et al., 2018].**
>
> We have updated the literature review to more clearly indicate how our work is differentiated from existing work, such as [Kumar et al., 2018]. To summarize the main differences to [Kumar et al., 2018] beyond the empirical improvements in Tables 1 and 2 of the attached PDF, MMCE is a special case of our framework applied to one-vs-all confidence calibration in classification. Our method enables enforcing many more forms of calibration (e.g., the 8 regression examples in Appendix A of the paper and the 3 classification examples in Table 3 of the attached PDF). One advantage of this added generality is that we can draw connections to decision making by designing calibration metrics specifically for a decision problem at hand, as in Section 5 of our paper. We will incorporate the comparisons to related work described in our general author response to the revised paper.
>
>
>
> > **What are the n and d in Table 2?**
>
> $n$ is the number of examples in the dataset and $d$ is the number of features. We will update the table caption to clarify this point.
>
>
>
> > **How to demonstrate the prediction performance improvement using the proposed methods?**
>
> We quantify prediction performance as the combination of calibration and sharpness. We define sharpness with the Negative Log Likelihood (NLL) and calibration with the Quantile Calibration Error (QCE). Note that the NLL is a strictly proper scoring rule so it can also be used as a measure of prediction performance on its own. In Table 2, we find that the best QCE is observed with our method in all 5 datasets, and the best NLL is observed with our method in 4 out of 5 datasets. Additionally, in the new classification experiments we report ECE, entropy, and accuracy (see Tables 1 and 2 in the PDF attached to this response). We find that our method outperforms the baseline NLL method and MMCE [Kumar et al., 2018] in the classification setting as measured by ECE, entropy, and accuracy.
>
>
>
> *[Kumar et al., 2018] Kumar, Aviral, Sunita Sarawagi, and Ujjwal Jain. Trainable calibration measures for neural networks from kernel mean embeddings. International Conference on Machine Learning. PMLR, 2018.*

---

> > ### Comment · Reviewer_UW9V · 2023-08-16
> > **After reading the rebuttal**
> >
> > Thanks for responding to my questions. Most of my concerns are addressed. I am happy to keep my score.

---

### Official Review · Reviewer_1Qie · 2023-07-05

**Soundness:** 3 good
**Presentation:** 3 good
**Contribution:** 3 good
**Rating:** 6
**Confidence:** 4

**Summary:**

This paper presents a general framework for train-time calibration based on kernel MMD. The authors demonstrate empirically that choosing good kernels (based on the task at hand) in their framework can lead to improvements in calibration while maintaining sharpness of predictions.

Update: I have increased my score to a 6 in light of the authors' responses. I think the empirical portions of the paper could still be improved (ideally comparing to more training-time calibration methods), but I believe the sharpness improvements over MMCE are interesting and useful.

**Strengths:**

- **Originality:** While kernel-based calibration methods have been studied in the literature, the perspective provided in this work is original in that the authors show how various different types of calibration can be captured by their MMD framework.
- **Quality:** The quality of the work is overall good, with appropriate support and examples provided for the claims in the paper.
- **Clarity:** The paper is mostly well-written. However, there are implementation details regarding the MMD approach that I feel are important and missing (detailed further in weaknesses).
- **Significance:** This paper adds to the growing body of work showing the benefits of kernel-based methods for improving calibration. However, it is difficult to assess the relative significance of this work (compared to, for example [1]) due to the limited experiments.

[1] https://proceedings.mlr.press/v80/kumar18a.html

**Weaknesses:**

- **Motivation for Implementation Details:** It would be very helpful to accompany the discussion in Section 4 with an idea of how the conditional kernel MMD is implemented in practice and why a practitioner would make certain decisions. For example, is using a model parameterization like the one in the paper actually that necessary (i.e. predicting parameters of a Gaussian)? Can one not just take a similar approach to [1]?
- **Experiments:** The experiments in the paper compare to a baseline of training using NLL (and post-hoc calibration), with no comparisons to other train-time calibration approaches. As a result, it is difficult to assess the relative worth of the proposed methodology compared to existing approaches.
- **Contextualization:** The MMD framework proposed in this paper is very similar to the ideas in [1], and I think the paper would benefit from more direct comparisons to [1] and related prior work that highlight the contributions of this work.

To summarize, I feel that the lack of implementation details + lack of comparison to other train-time calibration approaches limit the contribution of the paper.

**Questions:**

- How much longer does it take to train (wall clock time) using the MMD regularization?
- How important is the choice of output distribution for the model considered? In other words, did you consider predicting the parameters of a distribution other than Gaussian and comparing?
- As asked in the weaknesses, can we relax the need to differentiably sample from the predictive distribution?
- In the supplementary material, I believe under distribution calibration it should be $P_{Y \mid X}(y) \mid Q_{Y \mid X} = Q_{Y \mid X}(y)$.

**Limitations:**

While the authors do discuss limitations, I feel the limitations mentioned should have received further exposition in the main body of the work.

---

> ### Author Rebuttal · Authors · 2023-08-10
>
> Thank you for your thoughtful feedback on our work! We are grateful that you appreciated the quality and originality of our work. We believe we have thoroughly responded to each of your comments, and that these changes have improved our work. If you find our responses and the corresponding updates to our paper satisfactory, we would kindly request that you consider adjusting your score.
>
> > **It would be very helpful to accompany the discussion in Section 4 with an idea of how the conditional kernel MMD is implemented in practice and why a practitioner would make certain decisions. For example, is using a model parameterization like the one in the paper actually that necessary (i.e. predicting parameters of a Gaussian)?**
>
> Using a model that outputs the parameters of a Gaussian distribution is indeed not required. We chose to use Gaussian distributions in our experiments to show how our methods perform in a common setting for distribution prediction in regression. In new experiments and in our general author response, we have described how our methods apply to categorical distributions. What is required is that we can estimate and optimize the MMD expectations with respect to the forecasted distribution. In classification, we completely remove the need for sampling by computing the expectation with respect to the categorical distribution exactly. In regression, any reparameterizable distribution should work nicely. We will add a discussion of the model parameterization and design principles to our revised paper.
>
> >  **It is difficult to assess the relative worth of the proposed methodology compared to existing approaches.**
>
> We have added experiments comparing our methods to MMCE [Kumar et al, 2018] in the classification setting. See Tables 1 and 2 in the PDF attached to this response and the general response for additional details. We find that our objectives tend to produce better calibration, sharpness, and accuracy scores than MMCE both with and without post-hoc temperature scaling.
>
> > **The paper would benefit from more direct comparisons to [Kumar et al., 2018] and related prior work that highlight the contributions of this work.**
>
> In addition to the new empirical comparisons, we have expanded the related works section to include a more thorough comparison to [Kumar et al., 2018] and other existing work. See our general author response for a more direct comparison that highlights the contributions of our work.
>
> > **How much longer does it take to train (wall clock time) using the MMD regularization?**
>
> Compared to training with the NLL, our proposed method requires an average of 1.3x wall clock time on the five classification datasets, and 1.2x wall clock time for the five regression datasets. We will add these results to the updated paper.
>
> > **How important is the choice of output distribution for the model considered?**
>
> As mentioned above, it is not important that the model outputs a Gaussian distribution. In the new experiments with classification, we calibrate categorical distributions. We have found similar results when applying our methods to models that output Gaussian mixture distributions, using Gumbel Softmax relaxation to differentiate through the mixture distribution.
>
> > **Can we relax the need to differentiably sample from the predictive distribution?**
>
> We can relax the need for differentiable sampling in some cases. In the classification setting, there is no need to differentiably sample from the predictive distribution, since the MMD metric can be analytically computed (we implement this in the new classification experiments). In the regression setting, when a distribution does not admit differentiable sampling (e.g., a Mixture of Gaussians), we can apply a differentiable relaxation (e.g., using Gumbel-Softmax to differentiate with respect to the mixture weights). In practice, several highly expressive predictive distributions exist in the regression setting which can be trained using our method (e.g. Neural Autoregressive Flows [Huang et al, 2018]). In principle, it is also possible to remove the need for differentiable sampling in regression by optimizing our MMD objective using Monte-Carlo RL techniques (e.g. REINFORCE). This is an interesting direction for future research, but beyond the scope of this paper.
>
> > **In the supplementary material, I believe under distribution calibration it should be $P_{Y \mid X}(y) \mid Q_{Y \mid X} = Q_{Y \mid X}(y)$.**
>
> Thank you for catching the error, we have corrected it in the revised paper.
>
> *[Kumar et al., 2018] Kumar, Aviral, Sunita Sarawagi, and Ujjwal Jain. Trainable calibration measures for neural networks from kernel mean embeddings. International Conference on Machine Learning. PMLR, 2018.*
>
> *[Huang et al, 2018] Huang, Chin-Wei, et al. Neural autoregressive flows. International Conference on Machine Learning. PMLR, 2018.*

---

> > ### Comment · Reviewer_1Qie · 2023-08-17
> >
> > Thank you for the clarifications and the comparison to MMCE. For the comparison to MMCE, what exactly was the setup (i.e. did you follow their settings or use the same kernels as your models)? There seems to be some strange (possibly anomalous) behavior on the heart-disease dataset in which training with MMCE reduced accuracy and calibration significantly, which is surprising - everywhere else the performance of MMCE essentially matches that of the proposed method (the biggest ECE improvement is about 0.02, and this may not even be maintained across different ECE estimation procedures).
> >
> > My main question remains regarding the actual implementation details of the approach outlined in the paper - is a correct understanding that one chooses a kernel and a forecast distribution and then approximates the expectation in the MMD via sampling? If so, how many samples are used? I think in revising the paper it would be helpful to clearly outline the exact implementation details a practitioner would need to follow to implement the approach in practice.

---

> > > ### Author Response · Authors · 2023-08-18
> > >
> > > Thank you for engaging with our work! We truly believe that your comments and suggestions have helped us in strengthening our work and clarifying the experimental setup. We address your follow-up comments below.
> > >
> > > > **For the comparison to MMCE, what exactly was the setup (i.e. did you follow their settings or use the same kernels as your models)?**
> > >
> > > We adopted the same setup for MMCE as detailed in [Kumar et al., 2018]. Specifically, we utilized a Laplacian Kernel with a width of 0.4, and the $\lambda$ parameter for weighting MMCE with respect to NLL was chosen through a hyperparameter sweep. For MMD metrics, we followed the exact same setup detailed in Section 6.2 of our paper. For both MMCE and our models, we perform a hyperparameter sweep and select the best performing model configurations based on model accuracy on a held-out validation set. We retrain the best model configurations using 50 random seeds to obtain the final results presented in our tables.
> > >
> > > > **There seems to be some strange (possibly anomalous) behavior on the heart-disease dataset in which training with MMCE reduced accuracy and calibration significantly, which is surprising.**
> > >
> > > We ran additional hyperparameter sweeps to verify the results on the heart-disease dataset, and found the results to be consistent across runs. On the datasets we tested, we found that MMCE typically improved ECE and worsened entropy, relative to training with only NLL. We note that prior work has also found that MMCE worsens ECE and accuracy on some datasets. For example, on ResNet-110 [Mukhoti et al., 2021, Table 1] reports for MMCE that ECE=5.51 and error=5.4, as opposed to ECE=4.41 and error=4.89 when training with only cross-entropy. To further facilitate the reproducibility of our results, we plan to release our code.
> > >
> > > > **The performance of MMCE essentially matches that of the proposed method.**
> > >
> > > We would like to highlight that the main motivation for calibration regularizers is to mitigate the tradeoff between calibration and sharpness. We find that our method yields better sharpness than MMCE on every dataset we tested, often by a significant margin. MMCE gives entropy that ranges from 1.45x to 5.08x greater than ours w/o temp scaling, and 1.02x to 1.38x greater than ours w/ temp scaling (see Tables 1 & 2 in the pdf attached to our response). Essentially, compared to MMCE, our method achieves comparable (and often better) calibration without degrading sharpness.
> > >
> > > > **Is a correct understanding that one chooses a kernel and a forecast distribution and then approximates the expectation in the MMD via sampling? If so, how many samples are used?**
> > >
> > > Yes, your understanding is correct. To be clear, the metric is specified by a choice of kernel, target variable, forecast variable, and conditioning variable. To estimate the metric, we use sample estimates based on the dataset (e.g. ground truth $X$, $Y$) , and samples from the model’s forecasted distribution (e.g. sampled from $Q_{Y|X}$).
> > >
> > > For classification, we provide the expression for our estimator in the general response above (no sampling from the forecast is required). For the regression setting, we discuss the estimator of MMD in lines 124-128 of our paper. Although we mention that marginalizing randomness of the forecaster improves training stability in practice, we agree that this point can be easily missed, and will make sure to clarify it further in the revised paper. We found that increasing the number of samples from the forecasted distribution improves stability at the cost of additional compute. See the below table for quantitative results on the effect of the number of forecast samples on the NLL, QCE, and DCE. We find that model performance stabilizes at around 10 forecast samples per data example. This was the setting used in our experiments, and reflects the 1.2x wall clock training time relative to training with only NLL.
> > >
> > > Table values follow the format of Negative Log Likelihood (NLL) / Quantile Calibration Error (QCE) / Decision Calibration Error (DCE). All model hyperparameters were fixed between runs.
> > > | # samples | crime | blog | medical-expenditure | superconductivity | fb-comment |
> > > |:---:|:---:|:---:|:---:|:---:|:---:|
> > > | 1 | -0.703/ 0.198/ 0.088 | 0.952/ 0.379/ 4.171 | 1.546/ 0.071/ 0.448 | 3.369/ 0.053/ 0.182 | 0.637/ 0.268/ 3.150 |
> > > | 5 | -0.720/ 0.154/ 0.043 | 0.96/ 0.395/ 4.090 | 1.539/ 0.071/ 0.419 | **3.311**/ 0.038/ 0.216 | **0.409**/ 0.276/ 3.146 |
> > > | 10 | -0.777/ 0.154/ 0.060 | 0.85/ 0.416/ 4.077 | 1.54/ 0.072/ 0.459 | 3.333/ 0.036/ 0.208 | 0.590/ 0.278/ **3.102** |
> > > | 50 | -0.779/ **0.15**/ **0.041** | 0.847/ 0.432/ 4.016 | **1.531**/ 0.065/ 0.447 | 3.345/ 0.041/ 0.208 | 0.479/ **0.223**/ 3.140 |
> > > | 100 | -0.781/ 0.153/ 0.043 | **0.829**/ 0.386/ **3.978** | 1.536/ **0.064**/ 0.449 | 3.318/ 0.036/ **0.181** | 0.469/ 0.230/ 3.149 |
> > > | 200 | **-0.782**/ 0.151/ 0.043 | **0.829**/ **0.373**/ **3.978** | 1.535/ **0.064**/ 0.442 | 3.291/ **0.034**/ 0.182 | 0.458/ 0.231/ 3.137 |

---

> > > > ### Author Response · Authors · 2023-08-18
> > > > **References**
> > > >
> > > > *[Kumar et al., 2018] Kumar, A., Sunita S., and Ujjwal J. Trainable calibration measures for neural networks from kernel mean embeddings. ICML, 2018.*
> > > >
> > > > *[Mukhoti et al., 2020] Mukhoti, J., Kulharia, V., Sanyal, A., Golodetz, S., Torr, P., & Dokania, P. Calibrating deep neural networks using focal loss. In Neurips, 2020.*

---

> > > > ### Comment · Reviewer_1Qie · 2023-08-18
> > > >
> > > > Thank you very much for the further clarifications; I had glossed over the improvements in NLL relative to MMCE when I looked at the updated results. This is indeed a useful attribute of the proposed method, and I think in revising it would be helpful to emphasize this further (possibly with additional visualizations/investigations into sharpness vs calibration in the appendix). My only remaining question is regarding the NLL performance on the crime dataset - how is it negative? Sorry I missed this when doing an initial pass over the paper.

---

> > > > > ### Author Response · Authors · 2023-08-18
> > > > >
> > > > > We are glad you appreciate the contributions of our method for simultaneously achieving calibration and sharpness. Thank you for your suggestions! We will clarify and expand on the value of preserving sharpness in the revised paper.
> > > > >
> > > > > > My only remaining question is regarding the NLL performance on the crime dataset - how is it negative?
> > > > >
> > > > > The negative log-likelihood can be negative in a regression setting, since the probability density function can take values greater than one. For a concrete example, consider a forecaster $q$ that, given features $x$, outputs a Gaussian distribution $q_{Y \mid x} = \mathcal{N}(y; \mu(x), \sigma)$ with mean $\mu(x)$ and a fixed standard deviation $\sigma$. Suppose that the true label follows the forecasted Gaussian distribution $Y \sim q_{Y \mid x}$. The negative log-likelihood (NLL) is given by:
> > > > >
> > > > > $$
> > > > > NLL(D, q) = \mathbb{E}_ {(x,y) \sim D} \left[ - \log q_{Y \mid x} (y) \right]
> > > > > $$
> > > > >
> > > > > Then the NLL simplifies to:
> > > > >
> > > > > $$
> > > > > NLL(D, q) = \log \left( \sqrt{2 \pi \sigma^2} \right) + \frac{1}{2}
> > > > > $$
> > > > >
> > > > > For a small standard deviation $\sigma = 0.1$, the above equation evaluates to the negative value $NLL(D, q) = -0.884$.

---

> > > > > > ### Comment · Reviewer_1Qie · 2023-08-19
> > > > > >
> > > > > > Ah right of course, my apologies for missing that. I've upgraded my score to a 6. Thank you for the responses!

---

### Official Review · Reviewer_cmLZ · 2023-07-06

**Soundness:** 3 good
**Presentation:** 3 good
**Contribution:** 3 good
**Rating:** 6
**Confidence:** 3

**Summary:**

This paper introduces differentiable kernel-based calibration metrics that unify and generalize popular forms of calibration.
The notions of calibration are presented as distribution matching constraints by using various kernels in the Maximum Mean Discrepancy (MMD) metric.
Through empirical evaluation, it is demonstrated that incorporating these metrics as regularizers enhances calibration, sharpness and facilitates better decision-making in regression scenarios.


**Strengths:**

- The proposed metrics can be integrated in gradient based optimization alongside proper scoring rules.
- The paper extends on the discussion regarding the connection between calibration and decision making, which offers valuable insights for practical problems.
- The empirical results are sufficiently extensive and verify the effectiveness of integrating the proposed metrics into training objectives, as a means to mitigate calibration error.

**Weaknesses:**

The related work section does not sufficiently discuss prior works, and fails to highlight the specific advancements made by this paper in comparison. As the current paper proposes a kernel-based differentiable metric to be used as a regularizer alongside a proper scoring rule, there has to be a discussion of existing trainable calibration methods, especially kernel-based methods like [Kumar et al., 2018, Zhang et al., 2020, Popordanoska et al., 2022].


*[Zhang et al., 2020] Jize Zhang, Bhavya Kailkhura, and T Han. Mix-n-Match: Ensemble and compositional methods for uncertainty calibration in deep learning. In ICML 2020*

*[Popordanoska et al., 2022] Teodora Popordanoska, Raphael Sayer, Matthew B. Blaschko A Consistent and Differentiable Lp Canonical Calibration Error Estimator. In NeurIPS 2022*

**Questions:**

- The abstract claims that the proposed metric generalizes popular forms of calibration for classification and regression. Could you clarify the relationship of the metric with notions of calibration used in classification, i.e. top-label, marginal and canonical (see e.g. Eq. 1 2 and 3 in [Vaicenavicius et al., 2019, Kull et al., 2019]?
- What is the computational overhead of the proposed method?
- How is the bandwidth of the kernels chosen?

*[Vaicenavicius et al. 2019] Juozas Vaicenavicius, David Widmann, Carl Andersson, Fredrik Lindsten, Jacob Roll, Thomas B. Schön Evaluating model calibration in classification. In AISTATS 2019.*

*[Kull et al. 2019] Meelis Kull, Miquel Perello-Nieto, Markus Kängsepp, Telmo Silva Filho, Hao Song, Peter Flach Beyond temperature scaling: Obtaining well-calibrated multiclass probabilities with Dirichlet calibration. In NeurIPS 2019*

**Limitations:**

Limitations are discussed in section 7.

---

> ### Author Rebuttal · Authors · 2023-08-10
>
> Thank you for your helpful comments! We are glad you appreciated the trainable nature of our metrics, the decision making connections, and the experimental results. We agreed with your suggestions to expand on the applications to classification and to add more detailed descriptions of related work and we have incorporated them into the paper. Responses to your questions and comments can be found below.
>
> > **As the current paper proposes a kernel-based differentiable metric to be used as a regularizer alongside a proper scoring rule, there has to be a discussion of existing trainable calibration methods, especially kernel-based methods like [Kumar et al., 2018, Zhang et al., 2020, Popordanoska et al., 2022].**
>
> We have updated the related works section to clarify the specific advancements of our work, including with respect to [Kumar et al., 2018, Zhang et al., 2020, Popordanoska et al., 2022]. Please refer to our general author response for details about how our work relates to those works. The biggest differences surround the generality of our approach (the aforementioned methods all focus on a single form of calibration in classification) and the connections we show to decision making. In new experiments, we also find that our method performs favorably to MMCE [Kumar et al., 2018] in the multiclass classification setting, as measured by sharpness (entropy), calibration (ECE), and accuracy.
>
> > **Could you clarify the relationship of the metric with notions of calibration used in classification, i.e. top-label, marginal and canonical?**
>
> We have added details about how our framework enforces notions of calibration used in classification. See Table 3 in the PDF attached to this response for a description of how our framework incorporates top-label, marginal and canonical calibration in classification (labeled according to [Vaicenavicius et al., 2019] for ease of comparison). See Tables 1 and 2 in the PDF attached to this response for new experimental results in which our method performs favorably with respect to MMCE [Kumar et al., 2018] and temperature scaling without train-time regularization.
>
> > **What is the computational overhead of the proposed method?**
>
> Relative to training purely with the NLL objective, our proposed method requires an average wall clock time per epoch of 1.2x for the 5 regression datasets, and 1.3x for the 5 classification datasets. We will add these results to the updated paper.
>
> > **How is the bandwidth of the kernels chosen?**
>
> In our experiments, the bandwidth of the kernel is chosen to optimize quantile calibration on held-out data using hyperparameter sweeps. In general, a smaller kernel bandwidth corresponds to less smoothing and requires more data to reliably estimate the training objective.
>
> *[Kumar et al., 2018] Kumar, A., Sunita S., and Ujjwal J. Trainable calibration measures for neural networks from kernel mean embeddings. ICML, 2018.*
>
> *[Popordanoska et al., 2022] Popordanoska, T., Sayer, R., Blaschko M., A Consistent and Differentiable Lp Canonical Calibration Error Estimator. NeurIPS, 2022.*
>
> *[Zhang et al., 2020] Jize Zhang, Bhavya Kailkhura, and T Han. Mix-n-Match: Ensemble and compositional methods for uncertainty calibration in deep learning. ICML, 2020.*
>
> *[Vaicenavicius et al., 2019] Juozas Vaicenavicius, David Widmann, Carl Andersson, Fredrik Lindsten, Jacob Roll, Thomas B. Schön Evaluating model calibration in classification. AISTATS 2019.*

---

> > ### Comment · Reviewer_cmLZ · 2023-08-17
> > **Post rebuttal**
> >
> > I thank the authors for answering my questions and addressing my concerns. I would like to keep my original score.

---

### Official Review · Reviewer_hWT9 · 2023-07-09

**Soundness:** 3 good
**Presentation:** 2 fair
**Contribution:** 2 fair
**Rating:** 5
**Confidence:** 3

**Summary:**

Calibration as a training regularizer:

This paper focuses on investigating how to incorporate calibration metrics as training regularizers. It starts with summarizing different calibration metrics using a unified view from distribution matching. The paper then proposed a kernel-based general metric that maps labels and conditioning variables to an RKHS space and measures the maximum mean discrepancy on the kernel space. The paper then extends the metric to the decision-making space with two concrete examples. In experiments, combined with post-hoc calibration, using the proposed calibration kernel metric as a trainable regularizer can achieve a better evaluation of calibration in regression problems.

**Strengths:**

Pros:
The paper provides a unified view of many different calibration metrics.

The kernel-based calibration regularizer is novel.

The paper is relatively clearly written.

**Weaknesses:**

Cons:

1. The decision calibration seems to be a natural extension of the section 4, as the decision calibration framework has been proposed before. So the main contribution is based on the kernel-MMD-based calibration metric, which is relatively straightforward.

2. Also, the paper only suggests using the metric as a regularizer to incentivize decision calibration during training but does not really guarantee anything or analyze formally the impact on the decision calibration scenario.

3. The trainable regularizer needs to be combined with post-hoc calibration method, which may raise the question how effective the method is. Table 2 shows that the MMD alone does not always provide enough improvement.

**Questions:**

I may have missed it, but I am a little confused by the correspondence between the trainable calibration metric and the evaluation. Table 2 shows training using NLL and the proposed method (individual calibration) and evaluating using both quantile calibration and decision calibration. Why quantile calibration is used here? Why not individual calibration in the evaluation?

What happens if the problem is a classification problem? The evaluation of calibration is classification is usually tricky. How much the trainable regularize can benefit there?

Minor: table caption is usually on top of the table.

=====
After reading the rebuttal, I raised my score to 5 but still concerned about the lack of theoretical guarantees and the requirement of additional calibration data and step.

**Limitations:**

The paper discussed technical limitations.

---

> ### Author Rebuttal · Authors · 2023-08-10
>
> Thank you for your time and the helpful feedback! Below, we clarify our contributions surrounding decision calibration, the role of post-hoc calibration, and respond to additional questions. Given our response to the main issues you raised in the review, we kindly request a re-evaluation of your score, taking our rebuttal into consideration.
>
> > **The decision calibration framework has been proposed before. So the main contribution is based on the kernel-MMD-based calibration metric, which is relatively straightforward.**
>
> We would like to clarify some important differences between our work and the existing literature on decision calibration. Our framework allows us to introduce calibration objectives for decision problems that cannot be calibrated by the methods proposed by [Zhao et al., 2021] and [Sahoo et al., 2021], the two most similar works of which we are aware. Namely, we can enforce decision calibration for large, and sometimes infinite action spaces (see Example 1, line 200 for a simple concrete example). Additionally, [Zhao et al., 2021] and [Sahoo et al., 2021] each calibrate for a single notion of decision calibration ($L^k$ and threshold calibration respectively), while we show how one can regularize for a wide variety of decision problems (see, e.g., Examples 1 & 2 in our paper). Lastly, our work introduces training objectives for decision calibration, while [Zhao et al., 2021] and [Sahoo et al., 2021] provide post-hoc recalibration procedures. We will clarify the relationship to existing work on decision calibration in the revised paper.
>
> *[Zhao et al., 2021] Zhao, S., Kim, M., Sahoo, R., Ma, T., & Ermon, S. Calibrating predictions to decisions: A novel approach to multi-class calibration. Neurips, 2021.*
>
> *[Sahoo et al., 2021] Sahoo, R., Zhao, S., Chen, A., & Ermon, S. Reliable decisions with threshold calibration. Neurips, 2021.*
>
> > **The paper only suggests using the metric as a regularizer to incentivize decision calibration during training but does not really guarantee anything or analyze formally the impact on the decision calibration scenario.**
>
> Our metrics are optimized in expectation exactly when a forecaster is decision calibrated. When we estimate these metrics from data, we can draw on convergence results for MMD sample estimates. Assuming the MMD kernel is bounded by a value $K \geq 0$, the probability that the sample estimate for our metric on a dataset of size $n$ deviates its expectation by more than $t$ is bounded above by $\exp\left( \frac{-t^2}{16 K^2} \right)$ [Gretton et al., 2012]. Consequently, one can in principle link a sample estimate of our metric to guarantees on decision-making (e.g., accurate loss estimation and no regret decisions). However, the primary goal of our regularization approach is to mitigate the tradeoff between calibration and sharpness, not to provide calibration guarantees at training time. If strong calibration guarantees are needed, post-hoc calibration approaches are probably most appropriate (we can use both!). We find empirically that our methods have a positive impact on decision-making performance, and that this benefit is consistent whether or not post-hoc recalibration is performed (see Table 2 in our paper).
>
> > **The trainable regularizer needs to be combined with post-hoc calibration method, which may raise the question how effective the method is. Table 2 shows that the MMD alone does not always provide enough improvement.**
>
> Post-hoc recalibration is an effective strategy for achieving calibration, but usually comes at the cost of reduced sharpness. Regularization approaches to calibration aim to give a better trade-off between calibration and sharpness by jointly optimizing the two metrics. This intuition is supported by our experimental results, where the sharpness penalty for adding post-hoc recalibration is significantly reduced for models trained with our calibration regularization. These trends are supported by other works that regularize for calibration, such as [Kumar et al., 2018; Popordanoska et al., 2022]. Thus, we see post-hoc recalibration and calibration regularization as complementary, with regularization providing improved sharpness and post-hoc methods providing consistent calibration.
>
> > **Table 2 shows training using NLL and the proposed method (individual calibration) and evaluating using both quantile calibration and decision calibration. Why quantile calibration is used here? Why not individual calibration in the evaluation?**
>
> Our intent in reporting quantile calibration and decision calibration was to show that our method can simultaneously achieve strong performance in multiple forms of calibration. We used the individual calibration objective to train these models since this is a stringent constraint that we expect to improve a variety of forms of calibration (we do observe this in practice). We will add the individual calibration metric to the result tables for completeness.
>
> > **What happens if the problem is a classification problem? The evaluation of calibration is classification is usually tricky. How much the trainable regularize can benefit there?**
>
> We agree that this is interesting! We have added experiments applying our methods to five classification datasets. We find that regularizing with our training objectives improves sharpness (entropy), calibration (ECE), and accuracy relative to MMCE [Kumar et al., 2018] and temperature scaling (see Tables 1 & 2 in the attached PDF, and our general author response).
>
> > **Minor: table caption is usually on top of the table.**
>
> Thank you for catching this formatting issue, we have corrected it in the revised paper.
>
> *[Kumar et al., 2018] Kumar, A., Sunita S., and Ujjwal J. Trainable calibration measures for neural networks from kernel mean embeddings. ICML, 2018.*
>
> *[Popordanoska et al., 2022] Popordanoska, T., Sayer, R., Blaschko M., A Consistent and Differentiable Lp Canonical Calibration Error Estimator. NeurIPS, 2022.*

---

### Author Rebuttal · Authors · 2023-08-10

We thank the reviewers for engaging thoughtfully and providing constructive feedback. We are glad that all reviewers appreciate that this work is “novel” [hWT9], “original” [1Qie], “a very interesting view for uncertainty calibration” [UW9V], and “offers valuable insights for practical problems” [cmLZ].

Our main takeaways from the reviews were that we should (1) clarify the relationship to previous work, (2) add details on how our framework applies in the classification setting, and (3) add more experiments and implementation details for practical guidance. Based on these helpful comments, we have made the following updates to our paper:
1. We significantly expanded the literature review to clarify the relationship between our work and other kernel-based calibration methods.
2. We derived calibration metrics for classification in our framework (see attached PDF, Table 3) and provide estimators below. These estimators do not require differentiable sampling from the predicted distribution, and demonstrate that our methods apply beyond Gaussian prediction.
3. We performed new experiments on five classification datasets to compare our method to MMCE [Kumar et al., 2018] (see attached PDF, Tables 1 and 2). We found that our method performs favorably in terms of accuracy, calibration (ECE), and sharpness (entropy), both with and without post-hoc temperature scaling.

We expand on each of these updates in order below. We will carefully address all reviewer comments in the updated paper and we clarify some important questions in individualized responses to the reviewers.

1. **Contextualization with Existing Work**

Multiple reviewers requested clarity on how our work relates to existing kernel-based calibration methods. Below, we briefly discuss each paper raised by the reviewers, and clarify how our work is similar and distinct. In particular, we would like to emphasize generality (we show 11 concrete forms of calibration our framework can enforce) and applications to decision-making as areas that distinguish our work from existing works. We will add these discussions to the revised paper.

- *[Kumar et al., 2018] Kumar, Aviral, Sunita Sarawagi, and Ujjwal Jain. Trainable calibration measures for neural networks from kernel mean embeddings. International Conference on Machine Learning. PMLR, 2018.*

[Kumar et al., 2018] introduce MMCE, a kernel-based training objective to enforce a one-vs-all confidence calibration measure for multiclass classification. MMCE is a special case of our framework applied to top-label calibration (see attached PDF, Table 3). One key difference between our work and theirs is that we have identified 11 forms of calibration (8 regression, 3 classification) that fit into our framework, as opposed to the one form of classification calibration studied by [Kumar et al., 2018]. Empirically, we find that our methods outperform MMCE in terms of calibration, sharpness, and accuracy (see below).

- *[Popordanoska et al., 2022] Teodora Popordanoska, Raphael Sayer, Matthew B. Blaschko A Consistent and Differentiable Lp Canonical Calibration Error Estimator. In NeurIPS, 2022.*

[Popordanoska et al., 2022] propose a differentiable estimator for distribution calibration exclusively in the classification setting, via kernel density estimation (KDE). Key differences between our method and theirs include: our methods apply to regression problems, we enforce local and group calibration by incorporating conditioning variables (see our Figure 1), and we can tailor our metrics for particular decision problems. In the revised version of our paper, we will include an empirical comparison to [Popordanoska et al., 2022] in the classification setting.


- *[Zhang et al., 2020] Jize Zhang, Bhavya Kailkhura, and T Han. Mix-n-Match: Ensemble and compositional methods for uncertainty calibration in deep learning. In ICML, 2020.*

[Zhang et al., 2020] also consider kernel-based measures of miscalibration. They use kernel density estimates to give a smoothed estimate of the conditional label distribution given a particular prediction in multiclass classification. Our work differs from [Zhang et al., 2020] in that we consider regression problems, we introduce trainable metrics, and we introduce calibration metrics tailored to downstream decision problems.

2. **Enforcing Calibration in Classification**

See Table 3 of the attached PDF for a description of how our framework expresses calibration in classification tasks. Here, we show how to estimate the MMD metric in a classification setting by marginalizing out the randomness originating from the forecast. This allows us to compute our MMD metric without sampling from the predicted distribution.

Let the label space be $\mathcal{Y}=\\{1,2,\dots,m\\}$. The sample estimate for MMD is

$\widehat{\textrm{MMD}}(\mathcal{F},D,Q) =  \frac{1}{n(n-1)} \sum_{i=1}^{n} \sum_{j \ne i}^{n} h_{ij}$,

where

$$h_{ij} = \sum_{y\in\mathcal{Y}} \sum_{y' \in \mathcal{Y}} q_i(y)q_j(y')k((y,z_i), (y',z_j))-2 \sum_{y \in \mathcal{Y}} q_i(y)k((y,z_i), (y_j,z_j)) + k((y_i,z_i), (y_j,z_j))$$

Here, $q_i(y)$ is the predicted probability of the label $Y=y$ for example $i$. This estimate is differentiable with respect to the predicted label probabilities and can be computed in $O(n^2m^2)$ time, where $m$ is the number of classes and $n$ is the number of datapoints.

3. **New Classification Experiments**

We evaluate the effectiveness of our method on five commonly used UCI classification datasets. In Tables 1 and 2 in the attached PDF, we compare our method’s performance to NLL and MMCE [Kumar et al., 2018]. For model selection and training, we follow the same experimental setup detailed in Section 6.2 of the paper.

Empirically, we find that our method performs favorably in terms of accuracy, calibration (ECE), and sharpness (entropy). These advantages relative to MMCE are present both with and without post-hoc temperature scaling.

---

### Decision · Program_Chairs · 2023-09-21

**Decision:**

Accept (poster)

**Comment:**

This paper explores the family of training techniques which incorporate calibration metrics as training regularizers. They state a unified view of different calibration metrics, and then propose a kernel-MMD based metric. The proposal performs well experimentally in several regression settings.
Reviewers agree that the method is novel, although fairly similar to Kumar et al. 2018. Further, the paper is well-written, and the experiments are thorough. The authors have responded to many of the reviewer's concerns in the rebuttal. Thus, I recommend acceptance.
I encourage the authors to address the reviewers' comments in the camera-ready. In particular, the paper would be stronger with a clearer comparison to MMCE.